# FEDDRO: FEDERATED COMPOSITIONAL OPTIMIZATION FOR DISTRIBUTIONALLY ROBUST LEARNING

## ABSTRACT

Recently, compositional optimization (CO) has gained popularity because of its applications in distributionally robust optimization (DRO) and many other machine learning problems. Large-scale and distributed availability of data demands the development of efficient federated learning (FL) algorithms for solving CO problems. Developing FL algorithms for CO is particularly challenging because of the compositional nature of the objective. Moreover, current state-of-the-art methods to solve such problems rely on large batch gradients (depending on the solution accuracy) not feasible for most practical settings. To address these challenges, in this work, we propose efficient FedAvg-type algorithms for solving non-convex CO in the FL setting. We first establish that vanilla FedAvg is not suitable to solve distributed CO problems because of the data heterogeneity in the compositional objective at each client which leads to the amplification of bias in the local compositional gradient estimates. To this end, we propose a novel FL framework FedDRO that utilizes the DRO problem structure to design a communication strategy that allows FedAvg to control the bias in the estimation of the compositional gradient. A key novelty of our work is to develop solution accuracy-independent algorithms that do not require large batch gradients (and function evaluations) for solving federated CO problems. We establish $\mathcal{O}(\epsilon^{-2})$ sample and $\mathcal{O}(\epsilon^{-3/2})$ communication complexity in the FL setting while achieving linear speedup with the number of clients. We corroborate our theoretical findings with empirical studies on large-scale DRO problems.

## 1 INTRODUCTION

Compositional optimization (CO) problems deal with the minimization of the composition of functions. A standard CO problem takes the form

$$\min_{x \in \mathbb{R}^d} f(g(x)) \text{ with } g(x) := \mathbb{E}_{\zeta \sim \mathcal{D}_g}[g(x; \zeta)], \tag{1}$$

where $x \in \mathbb{R}^d$ is the optimization variable, $f : \mathbb{R}^{d_g} \to \mathbb{R}$ and $g : \mathbb{R}^d \to \mathbb{R}^{d_g}$ are smooth functions, and $\zeta \sim \mathcal{D}_g$ represents a stochastic sample of $g(\cdot)$ from distribution $\mathcal{D}_g$. CO finds applications in a broad range of machine learning applications, including but not limited to distributionally robust optimization (DRO) Qi et al. (2022), meta-learning Finn et al. (2017), phase retrieval Duchi & Ruan (2019), portfolio optimization Shapiro et al. (2021), and reinforcement learning Wang et al. (2017).

In this work, we focus on a more challenging version of the CO problem (1) that often arises in the DRO formulation Haddadpour et al. (2022). Specifically, the problems that jointly minimize the summation of a compositional and a non-compositional objective. DRO has recently garnered significant attention from the research community because of its capability of handling noisy labels Chen et al. (2022), training fair machine learning models Qi et al. (2022), imbalanced Qi et al. (2020a) and adversarial data Chen & Paschalidis (2018). A standard approach to solve DRO is to utilize primal-dual algorithms Nemirovski et al. (2009) that are inherently slow because of a large number of stochastic constraints. The CO formulation enables the development of faster (dual-free) primal-only DRO algorithms Haddadpour et al. (2022). The majority of existing works to solve CO problems consider a centralized setting wherein all the data samples are available on a single server. However, modern large-scale machine-learning applications are characterized by the distributed collection of data by multiple clients Kairouz et al. (2021). This necessitates the development of distributed algorithms to solve the DRO problem.

Federated learning (FL) is a distributed learning paradigm that allows clients to solve a joint problem in collaboration with a server while keeping the data of each client private McMahan et al. (2017). The clients act as computing units where within each communication round, the clients perform multiple updates while the server orchestrates the parameter sharing among clients. Numerous FL algorithms exist in the literature to tackle standard (non-compositional) optimization problems Li et al. (2019; 2020); Karimireddy et al. (2019); Sharma et al. (2019); Zhang et al. (2021); Khanduri et al. (2021); Karimireddy et al. (2020). However, there is a lack of efficient distributed implementations when it comes to CO problems. The major challenges in developing FL algorithms for solving the CO problem are:

[**C1**]: The compositional structure of the problem leads to *biased* stochastic gradient estimates and this bias is amplified during local updates, which makes the theoretical analysis of the gradient-based algorithms intractable Chen et al. (2021).

[**C2**]: Typically, data distribution at each client is different, referred to as data heterogeneity. Heterogeneously distributed compositional objective results in *client drift* during local updates that lead to divergence of federated CO algorithms. This is in sharp contrast to the standard FedAvg for non-CO objectives where client drift can be controlled during the local updates Karimireddy et al. (2019).

[**C3**]: A majority of algorithms for solving CO rely on accuracy-dependent large batch gradients where the batch size depends on the desired solution accuracy, which is not practical from an implementation point of view Huang et al. (2021); Haddadpour et al. (2022); Guo et al. (2022).

These challenges naturally lead to the following question:

> *Can we develop FL algorithms that tackle* [**C1**] − [**C3**] *to solve CO in a distributed setting?*

In this work, we address the above question and develop a novel FL algorithm to solve typical versions of the CO problem that arise in DRO (Section 2). The major contributions of our work are:

- We for the first time present a negative result that establishes that the vanilla FedAvg (customized to CO) is **incapable of solving** the CO problems as it leads to bias amplification during the local updates. This shows that additional communication/processing is required by FedAvg to mitigate the bias in the local gradient estimation.

- We develop FedDRO, a novel FL algorithm for solving problems with both **compositional and non-compositional non-convex objectives** at the same time. To the best of our knowledge, such an algorithm has been absent from the open literature so far. Importantly, FedDRO addresses the above-mentioned challenges by developing several key innovations in the algorithm design.
  - FedDRO addresses [**C1**] by designing a **communication strategy** that utilizes the specific problem structure resulting from the DRO formulation and allows us to control the gradient bias. Specifically, FedDRO utilizes the fact that the compositional functions $g(\cdot)$ are often **low-dimensional embeddings** in the DRO formulation (see Examples in Section 2.1) and can be shared without incurring significant communication costs.
  - To address [**C2**], we **design the local updates** at each client so that the client drift is bounded. Our analysis captures the effect of data heterogeneity on the performance of FedDRO.
  - To address [**C3**], we utilize a **hybrid momentum-based estimator** to learn the compositional embedding and combine it with a stochastic gradient (SG) estimator to conduct the local updates. This construction allows us to circumvent the need to compute large accuracy-dependent batch sizes for computing the gradients and the compositional function evaluations.

- We establish the **convergence** of FedDRO and show that to achieve an $\epsilon$-stationary point, FedDRO requires $\mathcal{O}(\epsilon^{-2})$ samples while achieving **linear speed-up** with the number of clients, i.e., requiring $\mathcal{O}(K^{-1}\epsilon^{-2})$ samples per client. Moreover, FedDRO requires sharing of $\mathcal{O}(\epsilon^{-3/2})$ high-dimensional parameters and $\mathcal{O}(K^{-1}\epsilon^{-2})$ low dimensional embeddings per client.

**Notations.** The expected value of a random variable (r.v) $X$ is denoted by $\mathbb{E}[X]$. Conditioned on an event $\mathcal{F}$ the expectation of a r.v $X$ is denoted by $\mathbb{E}[X|\mathcal{F}]$. We denote by $\mathbb{R}$ (resp. $\mathbb{R}^d$) the real line (resp. the $d$ dimensional Euclidean space). We denote by $[K] := \{1, \dots K\}$. The notation $\|\cdot\|$ defines a standard $\ell_2$-norm. For a set $B$, $|B|$ denotes the cardinality of $B$. We use $\xi \sim \mathcal{D}_h$ and $\zeta \sim \mathcal{D}_g$ to denote the stochastic samples of functions $h(\cdot)$ and $g(\cdot)$ from distributions $\mathcal{D}_h$ and $\mathcal{D}_g$, respectively. A batch of samples from $h(\cdot)$ (resp. $g(\cdot)$) is denoted by $b_h$ (resp. $b_g$). Moreover, joint samples of $h(\cdot)$ and $g(\cdot)$ are denoted by $\bar{\xi} = \{b_h, b_g\}$. We represent by $\bar{x}$ the empirical average of a sequence of vectors $\{x_k\}_{k=1}^K$.

## 2 PROBLEM

In this work, we focus on a general version of the CO problem defined in (1). We consider the following problem that often arises in DRO (see Section 2.1) in a distributed setting with $K$ clients

$$\inf_{x \in \mathbb{R}^d} \left\{ \Phi(x) := h(x) + f(g(x)) \right\} \text{ with } h(x) := \tfrac{1}{K} \sum_{k=1}^K h_k(x) \ \& \ g(x) := \tfrac{1}{K} \sum_{k=1}^K g_k(x), \quad (2)$$

where each client $k \in [K]$ has access to the local functions $h_k : \mathbb{R}^d \to \mathbb{R}$ and $g_k : \mathbb{R}^d \to \mathbb{R}^{d_g}$ while $f(\cdot)$ is same as (1). The local functions $h_k(\cdot)$ and $g_k(\cdot)$ at each client $k \in [K]$ are: $h_k(x) = \mathbb{E}_{\xi_k \sim \mathcal{D}_{h_k}}[h_k(x; \xi_k)]$ and $g_k(x) = \mathbb{E}_{\zeta_k \sim \mathcal{D}_{g_k}}[g_k(x; \zeta_k)]$ and where $\xi_k \sim \mathcal{D}_{h_k}$ (resp. $\zeta_k \sim \mathcal{D}_{g_k}$) represents a sample of $h_k(\cdot)$ (resp. $g_k(\cdot)$) from distribution $\mathcal{D}_{h_k}$ (resp. $\mathcal{D}_{g_k}$). Moreover, the data at each client is heterogeneous, i.e., $\mathcal{D}_{h_k} \neq \mathcal{D}_{h_\ell}$ and $\mathcal{D}_{g_k} \neq \mathcal{D}_{g_\ell}$ for $k \neq \ell$ and $k, \ell \in [K]$.

In comparison to the basic CO in (1), (2) is significantly challenging, first, because of the presence of both compositional and non-compositional objectives and second, because of the distributed nature of the compositional function $g(\cdot)$.

*Remark* 2.1 (Comparison to Gao et al. (2022) and Huang et al. (2021)). Note that formulation (2) is significantly different than the setting considered in Huang et al. (2021); Gao et al. (2022). Specifically, our formulation considers a practical setting where the compositional functions are distributed across agents, i.e., the function is $g = 1/K \sum_{k=1}^K g_k(x)$. In contrast, Huang et al. (2021); Gao et al. (2022) consider a setting with objective $\tfrac{1}{k} \sum_{k=1}^K f_k(g_k(\cdot))$, note here that the compositional function is local to each agent. This implies that algorithms developed in Huang et al. (2021); Gao et al. (2022) cannot solve problem (2). Importantly, problem (2) models realistic FL training settings while being more challenging compared to Huang et al. (2021); Gao et al. (2022) since in (2) the data heterogeneity of the inner problem also plays a role in the convergence of the FL algorithm. Please see the discussion in Appendix A.1 for more details.

### 2.1 EXAMPLES: CO REFORMULATION OF DRO PROBLEMS

In this section, we discuss different DRO formulations that can be efficiently solved using CO Haddadpour et al. (2022). DRO problem with a set of $m$ training samples denoted as $\{\zeta_i\}_{i=1}^m$ is

$$\min_{x \in \mathbb{R}^d} \max_{\mathbf{p} \in P_m} \sum_{i=1}^m p_i \ell(x; \zeta_i) - \lambda D_*(\mathbf{p}, \mathbf{1}/m) \quad (3)$$

where $x \in \mathbb{R}^d$ is the model parameter, $P_m := \{\mathbf{p} \in \mathbb{R}^m : \sum_{i=1}^m p_i = 1, p_i \geq 0\}$ is $m$-dimensional simplex, $D_*(\mathbf{p}, \mathbf{1}/m)$ is a divergence metric that measures distance between $\mathbf{p}$ and uniform probability $\mathbf{1}/m \in \mathbb{R}^m$, and $\ell(x, \zeta_i)$ denotes the loss on sample $\zeta_i$, $\rho$ is a constraint parameter, and $\lambda$ is a hyperparameter. Next, we discuss two popular reformulations of (3) in the form of CO problems.

**DRO with KL-Divergence.** Problem (3) is referred to as a KL-regularized DRO when the distance metric $D_*(\mathbf{p}, \mathbf{1}/m)$ is the KL-Divergence, i.e., we have $D_*(\mathbf{p}, \mathbf{1}/m) = D_{\mathrm{KL}}(\mathbf{p}, \mathbf{1}/m)$ with $D_{\mathrm{KL}}(\mathbf{p}, \mathbf{1}/m) := \sum_{i=1}^m p_i \log(p_i m)$. For this case, an equivalent reformulation of (3) is

$$\min_{x \in \mathbb{R}^d} \log \left( \tfrac{1}{m} \sum_{i=1}^m \exp\left( \tfrac{\ell(x; \zeta_i)}{\lambda} \right) \right), \quad (4)$$

which is a CO with $g(x) = 1/m \sum_{i=1}^m \exp(\ell(x; \zeta_i)/\lambda)$, $f(g(x)) = \log(g(x))$ and $h(x) = 0$.

**DRO with $\chi^2$- Divergence.** Similar to KL-regularized DRO, (3) is referred to as a $\chi^2$-regularized DRO when $D_*(\mathbf{p}, \mathbf{1}/m)$ is the $\chi^2$-Divergence, i.e., we have $D_*(\mathbf{p}, \mathbf{1}/m) = D_{\chi^2}(\mathbf{p}, \mathbf{1}/m)$ with $D_{\chi^2}(\mathbf{p}, \mathbf{1}/m) := m/2 \sum_{i=1}^m (p_i - 1/m)^2$. For this case, an equivalent reformulation of (3) is

$$\min_{x \in \mathbb{R}^d} -\tfrac{1}{2\lambda m} \sum_{i=1}^m \left( \ell(x; \zeta_i) \right)^2 + \tfrac{1}{2\lambda} \left( \tfrac{1}{m} \sum_{i=1}^m \ell(x; \zeta_i) \right)^2 \quad (5)$$

which is again a CO with $g(x) = 1/m \sum_{i=1}^m \ell(x; \zeta_i)$, $f(g(x)) = g(x)^2/2\lambda$ and $h(x) = -\tfrac{1}{2\lambda m} \sum_{i=1}^m \left( \ell(x; \zeta_i) \right)^2$.

Note that both (4) and (5) can be equivalently restated in the practical FL setting of (2) if the overall samples are shared across multiple clients with each client having access to a subset of samples.

**Related work.** Please see Table 1 for a comparison of current approaches to solve CO problems in distributed settings. For a detailed review of centralized and distributed non-convex CO and DRO problems, please see Appendix A. Here, we point out some drawbacks of the current approaches to solving federated CO problems:

Table 1: Comparison with the existing works. Here, CO-ND refers to CO with a non-distributed compositional part (see Remark 2.1). CO + Non-CO refers to problems with both CO and Non-CO objectives. VR refers to variance reduction. (I) and (O) refers to the inner and outer loop, respectively.
* Theoretical guarantees for GCIVR exist only for the finite sample setting with $m$ total network-wide samples.

| ALGORITHM | SETTING | UPDATE | BATCH-SIZES | CONVERGENCE |
|---|---|---|---|---|
| ComFedL Huang et al. (2021) | CO-ND | SGD | $\mathcal{O}(\epsilon^{-2})$ | $\mathcal{O}(\epsilon^{-4})$ |
| Local-SCGDM Gao et al. (2022) | CO-ND | Momentum SGD | $\mathcal{O}(1)$ | $\mathcal{O}(\epsilon^{-2})$ |
| FedNest Tarzanagh et al. (2022) | Bilevel | VR | $\mathcal{O}(1)$ | $\mathcal{O}(\epsilon^{-2})$ |
| GCIVR* Haddadpour et al. (2022) | CO + Non-CO | VR | $\sqrt{m}$ (I), $m$ (O) | $\mathcal{O}(\min\{\sqrt{m}\epsilon^{-1}, \epsilon^{-1.5}\})$ |
| FedDRO (Ours) | CO + Non-CO | SGD | $\mathcal{O}(1)$ | $\mathcal{O}(K^{-1}\epsilon^{-2})$ |

– None of the current works guarantee linear speedup with the number of clients Huang et al. (2021); Haddadpour et al. (2022); Tarzanagh et al. (2022); Gao et al. (2022).

– Utilize complicated multi-loop algorithms with momentum or VR-based updates Tarzanagh et al. (2022) that sometime require computation of large batch size gradients Haddadpour et al. (2022) to guarantee convergence. Such algorithms are not preferred in practical implementations.

– Consider a restricted setting where the compositional objective is not distributed among nodes Huang et al. (2021); Gao et al. (2022). Importantly, the algorithms developed therein cannot solve the problem considered in our work (see Appendix A.1).

Our work addresses all these issues and develops, FedDRO, the first simple SGD-based FL algorithm to tackle CO problems with the distributed compositional objective. Please see Table 1 for a comparison of the above works.

## 3 PRELIMINARIES

In this section, we introduce the assumptions, definitions, and preliminary lemmas.

**Definition 3.1** (Lipschitzness). For all $x_1, x_2 \in \mathbb{R}^d$, a differentiable function $\Phi : \mathbb{R}^d \to \mathbb{R}$ is: Lipschitz smooth if $\|\nabla\Phi(x_1) - \nabla\Phi(x_2)\| \leq L_\Phi\|x_1 - x_2\|$ for some $L_\Phi > 0$; Lipschitz if $\|\Phi(x_1) - \Phi(x_2)\| \leq B_\Phi\|x_1 - x_2\|$ for some $B_\Phi > 0$ and; Mean-Squared Lipschitz if $\mathbb{E}_\xi\|\Phi(x_1;\xi) - \Phi(x_2;\xi)\|^2 \leq B_\Phi^2\|x_1 - x_2\|^2$ for some $B_\Phi > 0$.

We make the following assumptions on the local and global functions in problem (2).

**Assumption 3.2** (Lipschitzness). The following holds

1. The functions $f(\cdot)$, $h_k(\cdot)$, $g_k(\cdot)$ for all $k \in [K]$ are differentiable and Lipschitz-smooth with constants $L_f, L_h, L_g > 0$, respectively.

2. The function $f(\cdot)$ is Lipschitz with constant $B_f > 0$ and $g_k(\cdot)$ is mean-squared Lipschitz for all $k \in [K]$ with constant $B_g > 0$.

Next, we introduce the variance and heterogeneity assumptions.

**Assumption 3.3** (Unbiased Gradient and Bounded Variance). The stochastic gradients and function evaluations of the local functions at each client are unbiased and have bounded variance, i.e.,

$$\mathbb{E}_{\xi_k}[\nabla h_k(x;\xi_k)] = \nabla h_k(x), \ \mathbb{E}_{\zeta_k}[\nabla g_k(x;\zeta_k)] = \nabla g_k(x), \ \mathbb{E}_{\zeta_k}[g_k(x;\zeta_k)] = g_k(x),$$
$$\mathbb{E}_{\zeta_k}[\nabla g_k(x;\zeta_k)\nabla f(y)] = \nabla g_k(x)\nabla f(y)$$

and
$$\mathbb{E}_{\xi_k}\|\nabla h_k(x;\xi_k) - \nabla h_k(x)\|^2 \leq \sigma_h^2,$$
$$\mathbb{E}_{\zeta_k}\|\nabla g_k(x;\zeta_k) - \nabla g_k(x)\|^2 \leq \sigma_g^2, \ \mathbb{E}_{\zeta_k}\|g_k(x;\zeta_k) - g_k(x)\|^2 \leq \sigma_g^2,$$

for some $\sigma_h, \sigma_g > 0$ and for all $x \in \mathbb{R}^d$ and $k \in [K]$.

**Assumption 3.4** (Bounded Heterogeneity). The heterogeneity $h_k(\cdot)$ and $g_k(\cdot)$ is characterized as

$$\sup_{x \in \mathbb{R}^d} \|\nabla h_k(x) - \nabla h(x)\|^2 \leq \Delta_h^2 \ \text{ and } \ \sup_{x \in \mathbb{R}^d} \|\nabla g_k(x) - \nabla g(x)\|^2 \leq \Delta_g^2,$$

for some $\Delta_h, \Delta_g > 0$ for all $k \in [K]$.

A few comments regarding the assumptions are in order. We note that the above assumptions are commonplace in the context of non-convex CO problems. Specifically, Assumption 3.2 is required to establish Lipschitz smoothness of the $\Phi(\cdot)$ (see Lemma 3.5) and is standard in the analyses of CO problems Wang et al. (2017); Chen et al. (2021). Assumption 3.3 captures the effect of stochasticity in the gradient and function evaluations of the CO problem while Assumption 3.4 characterizes the data heterogeneity among clients. We note that these assumptions are standard and have been utilized in the past to establish the convergence of many FL non-CO algorithms Yu et al. (2019a); Karimireddy et al. (2019); Khanduri et al. (2021); Zhang et al. (2021); Woodworth et al. (2020).

**Lemma 3.5** (Lipschitzness of $\Phi$). *Under Assumption 3.2 the compositional function, $\Phi(\cdot)$, defined in (2) is Lipschitz smooth with constant:* $L_\Phi := L_h + B_f L_g + B_g^2 L_f > 0.$

Lemma 3.5 establishes Lipschitz smoothness (Definition 3.1) of the compositional function $\Phi(\cdot)$. In general, $\Phi(\cdot)$ is a non-convex function, and therefore, we cannot expect to globally solve (2). We instead rely on finding approximate stationary points of $\Phi(\cdot)$ defined next.

**Definition 3.6** ($\epsilon$-stationary point). A point $x$ generated by a stochastic algorithm is an $\epsilon$-stationary point of a differentiable function $\Phi(\cdot)$ if $\mathbb{E}\|\nabla\Phi(x)\|^2 \leq \epsilon$, where the expectation is taken with respect to the stochasticity of the algorithm.

**Definition 3.7** (Sample and Communication Complexity). The sample complexity is defined as the total number of (stochastic) gradient and function evaluations required to achieve an $\epsilon$-stationary solution. Similarly, communication complexity is defined as the total communication rounds between the clients and the server required to achieve an $\epsilon$-stationary solution.

## 4 FEDERATED NON-CONVEX CO ALGORITHMS

In this section, we first establish the incapability of vanilla FedAvg to solve CO problems in general. Then, we design a communication-efficient FL algorithm to solve the non-convex CO problem.

### 4.1 CANDIDATE FEDAVG ALGORITHMS

---

**Algorithm 1** Vanilla FedAvg for non-convex CO

---

1: **Input**: Parameters: $\{\eta^t\}_{t=0}^{T-1}$, $I$
2: **Initialize**: $x_k^0 = \bar{x}^0, y_k^0 = \bar{y}^0$
3: **for** $t = 0$ to $T - 1$ **do**
4:     **for** $k = 1$ to $K$ **do**
5:       Update: $\begin{cases} \text{Compute } \nabla\Phi_k(x_k^t) \text{ using (6)} \\ x_k^{t+1} = x_k^t - \eta^t \nabla\Phi_k(x_k^t) \\ y_k^{t+1} = g_k(x_k^{t+1}) \end{cases}$
6:       **if** $t + 1 \mod I = 0$ **then**
7:         [Case I] Share: $\begin{cases} x_k^{t+1} = \bar{x}^{t+1} \end{cases}$
        [Case II] Share: $\begin{cases} x_k^{t+1} = \bar{x}^{t+1} \\ y_k^{t+1} = g_k(\bar{x}^{t+1}) \\ y_k^{t+1} = \bar{y}^{t+1} \end{cases}$
8:       **end if**
9:     **end for**
10: **end for**

---

In this section, we show that vanilla FedAvg is not suitable for solving federated CO problems of form (2). To establish this, we consider a simple deterministic setting with $h(x) = 0$. For this setting, the local gradients of $\Phi(\cdot)$ are estimated as

$$\nabla\Phi_k(x) = \nabla g_k(x_k)\nabla f(y_k), \quad (6)$$

where the sequence $y_k$ represents the local estimate of the inner function $g(x)$. To solve the above problem in a federated setup, we consider two candidate versions of FedAvg described in Case I and II of Algorithm 1. Similar to vanilla FedAvg, each agent performs multiple local updates within each communication round (see Step 5 of Algorithm 1). Moreover, since $g(x) := 1/k \sum_{k=1}^{k} g_k(x)$ with each agent $k \in [K]$ having access to only the local copy $g_k(\cdot)$, estimating $g(\cdot)$ locally within each communication round is not feasible. Therefore, each agent utilizes $y_k = g_k(x)$ as the local estimate of the inner function $g(\cdot)$. For communication, we consider two protocols. In the first setting, after $I$ local updates, in each communication round the agents share the locally updated parameters with the server and receive the aggregated parameter from the server (see Case I in Step 7). In the second setting, in addition to the locally updated parameters the agents also share their local function evaluations $y_k^t = g_k(x_k^t)$ with the server and receive the aggregated embedding $\bar{y}^t$ from the server. This step is utilized to improve the local estimates of $g(\cdot)$ (see Case II in Step 7). The algorithm executes for a total of $\lfloor T/I \rfloor$ communication rounds.

In the following, we show that Algorithm 1 is not a good choice to solve the federated CO problem presented in (2) even in the simple deterministic setting with $h(x) = 0$.

---

**Algorithm 2** Federated non-convex CO algorithm: FedDRO

1: **Input**: Parameters: $\{\beta^t\}_{t=0}^{T-1}, \{\eta^t\}_{t=0}^{T-1}, I$
2: **Initialize**: $x_k^{-1} = x_k^0 = \bar{x}^0, y_k^0 = \bar{y}^0$
3: **for** $t = 0$ to $T - 1$ **do**
4:     **for** $k = 1$ to $K$ **do**
5:         Sample $\bar{\xi}_k^t = \{b_{g_k}^t, b_{h_k}^t\}$ uniformly randomly from $\mathcal{D}_{g_k}$ and $\mathcal{D}_{h_k}$ respectively
6:         `Local Update and Sharing:` $\begin{cases} \text{Compute } y_k^t \text{ using (8) and share with the server} \\ \text{Receive } \bar{y}^t \text{ from the server and update } y_t^k = \bar{y}^t \\ \text{Compute } \nabla\Phi_k(x_k^t; \bar{\xi}_k^t) \text{ using (7)} \\ x_k^{t+1} = x_k^t - \eta^t \nabla\Phi_k(x_k^t; \bar{\xi}_k^t) \end{cases}$
7:         **if** $t + 1$ mod $I = 0$ **then**
8:             `Model Sharing:` $\left\{ x_k^{t+1} = \bar{x}^{t+1} \right.$
9:         **end if**
10:     **end for**
11: **end for**
12: **Return**: $\bar{x}^{a(T)}$ where $a(T) \sim \mathcal{U}\{1, ..., T\}$.

---

**Theorem 4.1** (Vanilla FedAvg: Non-Convergence for CO). *There exist functions $f(\cdot)$ and $g_k(\cdot)$ for $k \in [K]$ satisfying Assumptions 3.2, 3.3, and 3.4, and an initialization strategy such that for a fixed number of local updates $I > 1$, and for any $0 < \eta^t < C_\eta$ for $t \in \{0, 1, \dots, T-1\}$ where $C_\eta > 0$ is a constant, the iterates generated by Algorithm 1 under both Cases I and II do not converge to the stationary point of $\Phi(\cdot)$, where $\Phi(\cdot)$ is defined in (2) with $h(x) = 0$.*

Theorem 4.1 establishes that vanilla FedAvg is not suitable for solving federated CO problems. This naturally leads to the question of how can we modify FedAvg such that it can efficiently solve CO problems of the form (2)? Clearly, Theorem 4.1 suggests that sharing $y_k$'s in each iteration is required to ensure convergence of FedAvg since sharing the iterates $y_k$'s only intermittently leads to non-convergence of FedAvg. To this end, we propose to modify the FedAvg algorithm as presented in Algorithm 1 by sharing $y_k$ in each iteration $t \in \{0, 1, \dots, T-1\}$. The next result shows that the modified FedAvg resolves the non-convergence issue of FedAvg for solving CO problems.

**Theorem 4.2** (Modified FedAvg: Convergence for CO). *Suppose we modify Algorithm 1 such that $y_k^t = \bar{y}^t$ is updated at each iteration $t \in \{0, 1, \dots, T-1\}$ instead of $[t + 1 \mod I]$ iterations as in current version of Algorithm 1. Then if functions $f(\cdot)$ and $g_k(x)$ for $k \in [K]$ satisfy Assumptions 3.2, 3.3, and 3.4 such that for a fixed number of local updates $1 \le I \le \mathcal{O}(T^{1/4})$, there exists a choice of $\eta^t > 0$ for $t \in \{0, 1, \dots, T-1\}$ such that the iterates generated by (modified) Algorithm 1 converge to the stationary point of $\Phi(\cdot)$, where $\Phi(\cdot)$ is defined in (2) with $h(x) = 0$.*

Motivated by Theorem 4.2, we next develop a federated algorithm, FedDRO, to solve the problem (2) in a general stochastic setting with $h(x) \ne 0$.

## 4.2 FEDERATED NON-CONVEX CO ALGORITHM: FEDDRO

In this section, we propose a novel distributed non-convex CO algorithm, FedDRO, for solving (2). Note that as demonstrated in Section 4.1 this problem is particularly challenging because of the compositional structure of the problem combined with the fact that the data is heterogeneous for each client. Motivated by Theorem 4.2 above, in this work we develop a novel approach where we utilize the structure of the CO problem to develop efficient FL algorithms for solving (2). Specifically, as also demonstrated in Section 2.1 we utilize the fact that the embedding $g(\cdot)$ is a low-dimensional (e.g., $d_g = 1$) mapping, especially for the DRO problems. This implies that sharing of $g(\cdot)$ will be relatively cheap in contrast to the high-dimensional model parameters of size $d$ which can be very large and take values in millions or even in billions for modern overparameterized neural networks Vaswani et al. (2017). Therefore, like FedAvg, we share the model parameters intermittently after multiple local updates while sharing the low-dimensional embedding of $g(\cdot)$ frequently to handle the compositional objective. Moreover, to solve the CO problems for DRO the developed algorithms generally utilize batch sizes (for gradient/function evaluation) that are dependent on the solution accuracy Huang et al. (2021); Haddadpour et al. (2022). However, this is not feasible in most practical settings. In addition, to control the bias and to circumvent the need to compute large batch

gradients, we utilize a momentum-based estimator to learn the compositional function (see (8)) Chen et al. (2021). This construction allows us to develop FedAvg-type algorithms for solving non-convex CO problems wherein the local updates resemble the standard SGD updates.

The detailed steps of FedDRO are listed in Algorithm 2. During the local updates each client $k \in [K]$ updates its local model $x_k^t$ for all $t \in [T]$ using the local estimate of the stochastic gradients in Step 6. The local stochastic gradient estimates for each client $k \in [K]$ are denoted by $\nabla \Phi_k(x_k^t; \bar{\xi}_k)$ and are evaluated using the chain rule of differentiation as

$$\nabla \Phi_k(x_k^t; \bar{\xi}_k^t) = \frac{1}{|b_{h_k}^t|} \sum_{i \in b_{h_k}^t} \nabla h_k(x_k^t; \xi_{k,i}^t) + \frac{1}{|b_{g_k}^t|} \sum_{j \in b_{g_k}^t} \nabla g_k(x_k^t; \zeta_{k,j}^t) \nabla f(\bar{y}^t) \quad (7)$$

where $\bar{\xi}_k^t = \{b_{h_k}^t, b_{g_k}^t\}$ represents the stochasticity of the gradient estimate and $b_{h_k}^t = \{\xi_{k,i}^t\}_{i=1}^{|b_{h_k}^t|}$ (resp. $b_{g_k}^t = \{\zeta_{k,i}^t\}_{i=1}^{|b_{g_k}^t|}$) denotes the batch of stochastic samples of $h_k(\cdot)$ (resp. $g_k(\cdot)$) utilized to compute the stochastic gradient for each $k \in [K]$ and $t \in \{0, 1, \ldots, T-1\}$. The variable $\bar{y}^t$ is designed to estimate the inner function $1/K \sum_{k=1}^K g_k(x)$ in (2). A standard approach to estimate $g_k(x)$ locally for each $k \in [K]$ is to utilize a large batch such that the gradient bias from the inner function estimate can be controlled Guo et al. (2022); Huang et al. (2021); Haddadpour et al. (2022). In contrast, we adopt a momentum-based estimate of $g_k(\cdot)$ at each client $k \in [K]$ that leads to a small bias asymptotically Chen et al. (2021). We note that the estimator utilizes a hybrid estimator that combines a SARAH Nguyen et al. (2017) and SGD Ghadimi & Lan (2013) estimate for the function values rather than the gradients Cutkosky & Orabona (2019). Specifically, individual $y_k^t$'s are estimated in Step 6 as

$$y_k^t = (1 - \beta^t)\left(y_k^{t-1} - \frac{1}{|b_{g_k}^t|} \sum_{i \in b_t^{g_k}} g_k(x_k^{t-1}; \zeta_{k,i}^t)\right) + \frac{1}{|b_{g_k}^t|} \sum_{i \in b_{g_k}^t} g_k(x_k^t; \zeta_{k,i}^t). \quad (8)$$

for all $k \in [K]$ and where $\beta^t \in (0, 1)$ is the momentum parameter. Motivated by the discussion in Section 4.1, the parameters $y_k^t \in \mathbb{R}^{d_g}$ are shared with the server after the $y_k^t$ update, however, this sharing will not incur a significant communication cost since $y_k^t$'s are usually low dimensional embeddings (often a scalar with $d_g = 1$) as illustrated in Section 2.1 for DRO problems. The model parameters are then updated using the SG evaluated using (7). Finally, after $I$ local updates the model potentially high-dimensional model parameters are aggregated at the server and broadcasted back to the clients after aggregation in Step 8. Next, we state the convergence guarantees.

## 5 MAIN RESULT: CONVERGENCE OF FEDDRO

In the next theorem, we first state the main result of the paper detailing the convergence of FedDRO.

**Theorem 5.1** (Convergence of FedDRO). *For Algorithm 2, choosing the step-size $\eta^t = \eta = \sqrt{|b|K/T}$ and the momentum parameter $\beta^t = 4B_g^4 L_f^2 \cdot \eta^t$ for all $t \in \{0, 1, \ldots, T-1\}$. Moreover, with the selection of batch sizes $|b_{h_k}^t| = |b_{g_k}^t| = |b|$ for all $t \in \{0, 1, \ldots, T-1\}$ and $k \in [K]$, and for $T \geq T_{th}$ where $T_{th}$ is defined in Appendix F, then under Assumptions 3.2, 3.3 and 3.4 for $\bar{x}^{a(T)}$ chosen According to Algorithm 2, we have*

$$\mathbb{E}\left\|\nabla \Phi(\bar{x}^{a(T)})\right\|^2 \leq \underbrace{\frac{2\left[\Phi(\bar{x}^0) - \Phi(x^*) + \left\|\bar{y}^0 - g(\bar{x}^0)\right\|^2\right]}{\sqrt{|b|KT}}}_{Initialization}$$

$$+ \mathcal{C}(|b|, K, T, I)\underbrace{\left[C_{\sigma_h}\sigma_h^2 + C_{\sigma_g}\sigma_g^2\right]}_{Variance} + \mathcal{C}(|b|, K, T, I)\underbrace{\left[C_{\Delta_h}\Delta_h^2 + C_{\Delta_g}\Delta_g^2\right]}_{Heterogeneity},$$

*where $\mathcal{C}(|b|, K, T, I) := \max\left\{\frac{|b|K(I-1)^2}{T}, \frac{1}{\sqrt{|b|KT}}\right\}$ and constants $C_{\sigma_h}$, $C_{\sigma_g}$, $C_{\Delta_h}$, and $C_{\Delta_g}$ are defined in Appendix F*

We note that the condition on $T \geq T_{th}$ is required for theoretical purposes. Specifically, it ensures that the step-size $\eta = \sqrt{|b|K/T}$ is upper-bounded. A similar requirement has also been posed in Yu et al. (2019a;b); Khanduri et al. (2021) in the past. Theorem 5.1 captures the effect of heterogeneity, stochastic variance, and the initialization on the performance of FedDRO. As can be seen from the

expression in Theorem 5.1 the heterogeneity degrades the performance when the local updates, $I$, increase beyond a threshold, i.e., when the term $|b|K(I-1)^2/T$ dominates $1/\sqrt{|b|KT}$. The next result characterizes the possible choices of $I$ that ensure the efficient convergence of FedDRO.

**Corollary 5.2** (Local Updates). *Under the setting of Theorem 5.1 and choosing the number of local updates, $I$, such that we have $I \leq \mathcal{O}(T^{1/4}/(|b|K)^{3/4})$, the iterate $\bar{x}^{a(T)}$ chosen according to Algorithm 2 satisfies*

$$\mathbb{E}\big\|\nabla\Phi(\bar{x}^{a(T)})\big\|^2 \leq \underbrace{\frac{2\big[\Phi(\bar{x}^0) - \Phi(x^*) + \big\|\bar{y}^0 - g(\bar{x}^0)\big\|^2\big]}{\sqrt{|b|KT}}}_{Initialization} + \underbrace{\frac{C_{\sigma_h}\sigma_h^2 + C_{\sigma_g}\sigma_g^2}{\sqrt{|b|KT}}}_{Variance} + \underbrace{\frac{C_{\Delta_h}\Delta_h^2 + C_{\Delta_g}\Delta_g^2}{\sqrt{|b|KT}}}_{Heterogeneity}.$$

Corollary 5.2 states that there exists a choice of the number of local updates that guarantee that FedDRO achieves the same convergence performance as a standard FedAvg Karimireddy et al. (2019); Woodworth et al. (2020); Yu et al. (2019a); Khanduri et al. (2021) for solving the non-CO problems. Next, we characterize the sample and communication complexities of FedDRO.

**Corollary 5.3** (Sample and Communication Complexities). *Under the setting of Theorem 5.1 and choosing the number of local updates as $I = \mathcal{O}(T^{1/4}/(|b|K)^{3/4})$ the following holds*

*(i) The **sample complexity** of FedDRO is $\mathcal{O}(\epsilon^{-2})$. This implies that each client requires $\mathcal{O}(K^{-1}\epsilon^{-2})$ samples to reach an $\epsilon$-stationary point achieving linear speed-up.*

*(ii) The **communication complexity** of FedDRO is $O(\epsilon^{-3/2})$.*

The sample and communication complexities guaranteed by Corollary 5 match that of the standard FedAvg Yu et al. (2019b) for solving stochastic non-convex non-CO problems. We note that in addition to the $O(\epsilon^{-3/2})$ communication complexity that measures the sharing of high-dimensional parameters, FedDRO also shares $\mathcal{O}(K^{-1}\epsilon^{-2})$ low-dimensional embeddings (usually scalar values as illustrated in Section 2.1). Therefore, the total real values shared by each client during the execution of FedDRO is $\mathcal{O}(\epsilon^{-3/2}d + K^{-1}\epsilon^{-2})$. Notice that for high-dimensional models like training (large) neural networks, we will usually have $dK \geq \mathcal{O}(\epsilon^{-0.5})$ meaning the total communication will be $\mathcal{O}(\epsilon^{-3/2}d)$ which is better than any Federated CO algorithm proposed in the literature Huang et al. (2021); Gao et al. (2022); Guo et al. (2022). Importantly, to our knowledge this is the first work that ensures linear speed up in a federated CO setting, moreover, FedDRO achieves this performance without relying on the computation of large batch sizes.

## 6 EXPERIMENTS

In this section, we evaluate the performance of FedDRO with both centralized and distributed baselines. We, a) establish the superior performance of FedDRO in terms of training/testing accuracy, and b) evaluate the performance of FedDRO with different numbers of local updates to capture the effect of data heterogeneity. To evaluate the performance of FedDRO, we focus on two tasks: classification with an imbalanced dataset and learning with fairness constraints. For the first task, we use CIFAR10-ST and CIFAIR-100-ST datasets Qi et al. (2020b) (unbalanced versions of CIFAR10 and CIFAR100 Krizhevsky et al. (2009)) for image classification, and the performance is measured by training and testing accuracy achieved by different algorithms. For the second task, we use the Adult dataset Dua & Graff (2017) for enforcing equality of opportunity (on protected classes) on tabular data classification Hardt et al. (2016). For this setting, the performance is evaluated by training/testing accuracy, and the constraint violations, which are measured by the gap between the true positive rate of the overall data and the protected groups Haddadpour et al. (2022). Please see Appendix B for a detailed discussion of the classification problem, dataset description, experiment settings, and additional experimental evaluation.

**Baseline methods.** For the CIFAR10-ST and CIFAR100-ST datasets we compare FedDRO with popular centralized baselines for classification with imbalanced data. The baselines adopted for comparison are a popular DRO method, FastDRO Levy et al. (2020), a primal-dual SGD approach to solve constrained problems with many constraints, PDSGD Xu (2020), and a popular baseline minibatch SGD, MBSGD, customized for CO Ghadimi & Lan (2013). For the adult dataset, we use GCIVR Haddadpour et al. (2022) as the baseline distributed model to compare with FedDRO, since like FedDRO it is the only algorithm that can deal with compositional and non-compositional objectives at the same time. We also implement a simple parallel SGD as a baseline that ignores the fairness constraints, referred to as unconstrained in the experiments.

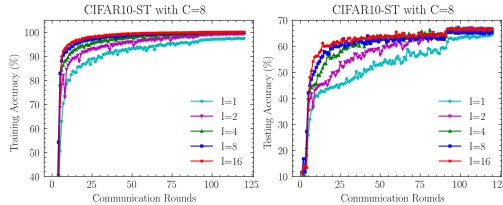

Figure 1: Train and test accuracy vs communication rounds for CIFAR10-ST and CIFAR100-ST.

**Implementation details.** We use 8 clients to model the distributed setting and split the (unbalanced) dataset equally for each client. We use ResNet20 for classification tasks on CIFAR10-ST and CIFAR100-ST datasets. For a fair comparison with centralized baselines, we choose $I = 1$ for FedDRO and implement a parallel version of the centralized algorithms where the overall gradient computation is $K$ times larger for each algorithm. This is to make sure that the overall gradient com-

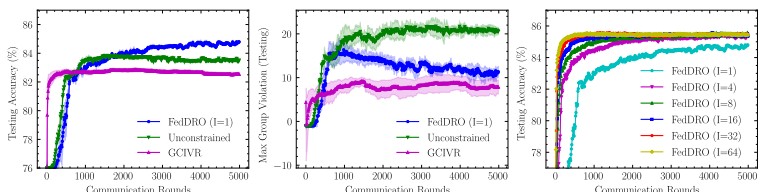

Figure 2: Train and test accuracy of FedDRO on the CIFAR10-ST and CIFAR100-ST for different $I$.

putations in each step are uniform across all algorithms. Performance with different values of $I$ is evaluated separately. For each algorithm, we used a batch size of 16 per client, and the learning rates were tuned from the set $\{0.001, 0.01, 0.05, 0.1\}$, the learning rate was dropped to $1/10^{\text{th}}$ after 90 communication rounds. For fairness-constrained classification on the Adult dataset, we use a logistic regression model. For this experiment, we adopt the parameter settings suggested in Haddadpour et al. (2022), for FedDRO we keep the same setting as in the earlier task. All results are averaged over 5 independent runs.

**Discussion.** In Figure 1, we evaluate the performance of Fed-DRO against the parallel implementations of the centralized baselines on unbalanced CIFAR datasets. Note that Fed-DRO provides superior training and comparable test accuracy to the

Figure 3: Comparison of FedDRO, GCIVR, and the unconstrained baseline (first two figures). Performance of FedDRO with different $I$ (rightmost figure).

state-of-the-art methods. In Figure 2, we evaluate the performance of FedDRO for a different number of local updates, $I$. Note that as $I$ increases the performance improves, however, beyond a certain, $I$, the performance doesn't improve capturing the effect of client drift because of data heterogeneity. Finally, in Figure 3 we assess the test performance of FedDRO against the distributed baseline GCIVR on the Adult dataset. We observe that FedDRO outperforms both GCIVR and unconstrained formulation in terms of accuracy and matches the constraint violation performance of GCIVR as communication rounds increase. Finally, for the rightmost image we evaluate the performance of FedDRO with different values of $I$, we notice that increasing the value of $I$ leads to improved performance, however, beyond a certain threshold (approximately over 32), the performance saturates as a consequence of client drift.

**Conclusion and limitations.** In this work, we first established that vanilla FedAvg algorithms are incapable of solving CO problems in the FL setting. To address this challenge, we showed that additional (low-dimensional) embeddings of the stochastic compositional objective are required to be shared to guarantee convergence of the SGD-based FL algorithms to solve CO of the form (2). To this end, we proposed FedDRO, the first federated CO framework that achieves linear speedup with the number of clients without requiring the computation of large batch sizes. We conducted numerical experiments on various real data sets to show the superior performance of FedDRO compared to state-of-the-art. An interesting future problem to be addressed includes limiting the privacy leakage of FedDRO while sharing the low-dimensional embeddings.

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
