APPENDIX

## A    RELATED WORK

**Centralized CO.** The first non-asymptotic analysis of stochastic CO problems was performed in Wang et al. (2017) where the authors proposed SCGD a two-timescale algorithm for solving problem (1). The convergence of SCGD was improved in Wang et al. (2016) where the authors proposed an accelerated variant of SCGD. Both SCGD and its accelerated variant achieved convergence rates strictly worse than SGD for solving non-CO problems. Recently, Ghadimi et al. (2020) and Chen et al. (2021) developed a single time-scale algorithm for solving the CO problem that achieves the same convergence as SGD for solving non-CO problems. Variance-reduced algorithms for solving the CO problems have also been considered in the literature, however, a major drawback of such approaches is the reliance of batch size on the desired solution accuracy Lian et al. (2017); Zhang & Xiao (2019); Hu et al. (2019).

**Distributed CO.** There have been only a few attempts to solve non-convex CO problems in the FL setting, partially, because of the challenges discussed in Section 1. The first FL algorithm to solve the non-convex CO problem, Compositional Federated Learning (ComFedL), was developed in Huang et al. (2021). ComFedL required accuracy-dependent batch sizes that resulted in $\mathcal{O}(\epsilon^{-4})$ convergence which is significantly worse compared to FedAvg to solve standard non-compositional problems Yu et al. (2019b). In Gao et al. (2022), Local Stochastic Compositional Gradient Descent with Momentum (Local-SCGDM) was proposed which removed the requirement of large batch sizes and achieved an $\mathcal{O}(\epsilon^{-2})$ convergence. However, Local-SCGDM utilized a non-standard momentum-based update from Ghadimi et al. (2020) that does not resemble a simple SGD-based update. Importantly, the CO problem solved by ComFedL Huang et al. (2021) and Local-SCGDM Gao et al. (2022) is non-standard as the problem is not distributed in the compositional objective (see Remark 2.1). In contrast, we consider a general setting where the compositional objective is also distributed among multiple nodes. Recently, Tarzanagh et al. (2022) proposed a nested optimization framework, FedNest, to solve bilevel problems in the FL setting. The proposed algorithm achieved SGD rates of $\mathcal{O}(\epsilon^{-2})$ Ghadimi & Lan (2013). Different from the simple SGD-based update rule, FedNest adopted a multi-loop variance reduction-based update. In Haddadpour et al. (2022), the authors proposed a Generalized Composite Incremental Variance Reduction (GCIVR) framework for solving problems of the form (2) in a distributed setting. GICVR achieved a better convergence rate of $\mathcal{O}(\epsilon^{-1.5})$, however, it relied on a double-loop structure and accuracy-dependent large batch sizes to achieve variance reduction. Importantly, none of the above works guarantee linear speedup with the number of clients. Moreover, the current algorithms utilize complicated momentum or VR-based update rules that require computation of accuracy-dependent batch sizes Haddadpour et al. (2022), and/or consider a simple setting where the compositional objective is not distributed among nodes Huang et al. (2021); Gao et al. (2022).

In contrast to all the above works, our work considers a general setting (2), where the goal is to jointly minimize a compositional and a non-compositional objective in the FL setting. To solve (2), we develop FedDRO a FedAvg algorithm for CO problems that achieves (i). the same guarantees as FedAvg for minimizing non-CO problems, (ii). linear speed-up with the number of clients, (iii). improved communication complexity, (iv). performance guarantees where the batch sizes required are independent of the desired solution accuracy, and (v). characterizes the performance as a function of local updates at each client and the data heterogeneity in the inner and outer non-compositional objectives.

**DRO.** DRO has been extensively studied in optimization, machine learning, and statistics literature Ben-Tal et al. (2013); Bertsimas et al. (2018); Duchi et al. (2021); Namkoong & Duchi (2017); Staib & Jegelka (2019) Broadly, DRO problem formulation can be divided into two classes, one is a constrained formulation and the other is the regularized formulation (see (3)) Levy et al. (2020); Duchi et al. (2021). A popular approach to solve the constrained DRO formulation is via primal-dual formulation where algorithms developed for min-max problems can directly be applied to solve constrained DRO Yan et al. (2019); Namkoong & Duchi (2017); Song et al. (2021); Alacaoglu et al. (2022); Tran Dinh et al. (2020). Many algorithms under different settings, e.g., convex, non-convex losses, and stochastic settings have been considered in the past to address such problems. However, primal-dual algorithms suffer from computational bottlenecks, since they require maintaining and updating the set of dual variables equal to the size of the dataset which can become particu-

larly challenging, especially for large-scale machine learning tasks. Recently, Levy et al. (2020) Qi et al. (2022) Haddadpour et al. (2022) have developed algorithms that are applicable to large-scale stochastic settings. Works Levy et al. (2020) and Qi et al. (2022) consider specific formulations of the DRO problem while Haddadpour et al. (2022) considers a general formulation, however, as pointed out earlier the algorithms developed in Haddadpour et al. (2022) are double loop and require accuracy-dependent batch sizes to guarantee convergence (see Table 1). In contrast, in this work, we develop algorithms that solve general instants of CO problems that often arise in DRO formulation. Importantly, the developed algorithms are amenable to large-scale distributed implementation with algorithmic guarantees independent of accuracy-dependent batch sizes.

### A.1 DETAILED COMPARISON WITH HUANG ET AL. (2021); GAO ET AL. (2022); TARZANAGH ET AL. (2022)

**Comparison with Huang et al. (2021); Gao et al. (2022).** We note that the problem setting in Huang et al. (2021) and Gao et al. (2022) is significantly different from the one considered in our work. We also would like to point out that the problem formulation considered in our work is more challenging than Huang et al. (2021); Gao et al. (2022) and the algorithms developed for solving the problem in Huang et al. (2021); Gao et al. (2022) cannot solve the problem considered in our work. In the following, we elaborate on the differences between our work and that of Huang et al. (2021); Gao et al. (2022).

In Huang et al. (2021); Gao et al. (2022), the authors consider the objective function

$$\frac{1}{k}\sum_{k=1}^{K} f_k(g_k(\cdot)). \tag{9}$$

Please observe that in this setting the local nodes have access to local composite functions $f_k(g_k(\cdot))$. In contrast, we consider a setting with objective function defined in (2) where the local nodes have access to only $h_k(\cdot)$ and $g_k(\cdot)$[1]. Note that the major difference in the two settings in (9) and (2) comes from the fact that in (9) the inner function $g_k(\cdot)$ is fully available at each node, whereas in (2) the inner function $1/K\sum_{k=1}^{K} g_k(\cdot)$ is not available (since each node can only access $g_k(\cdot)$) at the local nodes. Below, we discuss two major consequences of this:

- **Practicality:** We point out that the setting in (2) is more practical as can be seen from the examples presented in Section 2.1 wherein the DRO problems take the form of (2) rather than (9) in a distributed setting. For illustration, let us consider a simple setting where we have a total of $m$ samples with each node having access to $m_k = m/K$ samples. Then the DRO problem with KL-Divergence problem becomes

$$\min_{x \in \mathbb{R}^d} f\left(\frac{1}{k}\sum_{k=1}^{K} g_k(\cdot)\right) := \log\left(\frac{1}{m}\sum_{i=1}^{m} \exp\left(\frac{\ell_i(x)}{\lambda}\right)\right),$$

  where $f(\cdot) = \log(\cdot)$, $g_k(x) = 1/m_k \sum_{i=1}^{m_k} \exp\left(\ell_i(x)/\lambda\right)$, and $g(\cdot) = 1/K\sum_{k=1}^{K} g_k(\cdot)$. Note that the above formulation is same as (2) and cannot be formulated using (9). To demonstrate this fact we have used the notation in Table 1 as CO-ND for formulation of (9 where the inner function $g_k(\cdot)$ can be fully locally accessed by each node whereas our setting is more general with each node having only partial access to the inner-function $g(\cdot)$. Next, we show why the algorithms developed for Huang et al. (2021); Gao et al. (2022) cannot be utilized to solve the problem considered in our work.

- **Challenges in solving (2):** A major contribution of our work is in establishing the fact that the algorithms that are developed for solving 2), i.e., the algorithms developed in Huang et al. (2021); Gao et al. (2022), cannot be utilized to solve the problem considered in our work.

  To demonstrate this consider the simple deterministic setting with $f_k = f$, then the local gradient computed for the objective function in (2) will be $\nabla g_k(x) \nabla f(g_k(x))$ (please see (6) in the manuscript). Note that this is an unbiased local gradient for objective in (9) which further implies

---

[1] We would also like to note that the setting considered in the paper can be easily extended to the case where $f(\cdot) = 1/K\sum_{k=1}^{K} f_k(\cdot)$ without changing the current results.

that simple FedAVG-based implementations can be developed for solving this problem as done in Huang et al. (2021); Gao et al. (2022). In contrast, note that the local gradient $\nabla g_k(x)\nabla f(g_k(x))$ will be a biased local gradient for our problem in (2) and will lead to divergence of FedAvg-based algorithms Huang et al. (2021); Gao et al. (2022) as shown in Section 4.1. Moreover, note that we establish that even if we share the local functions $g_k(\cdot)$ intermitteltly among nodes we may not be able to mitigate the bias of local gradient and the developed algorithms will again diverge to incorrect solutions. Please see Section 4.1 for more details.

**Comparison with Tarzanagh et al. (2022).** Next, we note that the algorithm deveoped in Tarzanagh et al. (2022) is a bilevel algorithm with multi-loop structure with many tunable (hyper) parameters. Such algorithms are not preferred in practical implementations. In contrast our algorithm is a single-loop algorithm with simple FedAvg-type SGD updates. In addition to being practical, our work also significantly improves upon the theoretical guarantees achieved in Tarzanagh et al. (2022) by achieving linear speed-up with the number of clients as well as improved communication complexity which any of the works including Huang et al. (2021); Gao et al. (2022); Tarzanagh et al. (2022) are unable to achieve.

## B  DETAILED EXPERIMENT SETUP AND ADDITIONAL EXPERIMENTS

**Experiment setup.** The models are trained on an NVIDIA GeForce RTX 3090 GPU with 24 GB of memory. All experiments are conducted using the PyTorch framework, specifically Python 3.9.16 and PyTorch 1.8

**Datasets.** To evaluate the performance of FedDRO , the first section of the experiments is conducted on CIFAR10-ST and CIFAR-100-ST datasets for image classification. The second section of the experiments focuses on the Adult dataset, utilizing tabular data classification and emphasizing DRO for fairness constraints. The CIFAR10-ST and CIFAR-100-ST datasets are modified versions of the original CIFAR10 and CIFAR-100 datasets. The modification involves intentionally creating imbalanced training data. Specifically, only the last 100 images are retained for each class in the first half of the classes, while the other classes and the test data remain unchanged. This creates an imbalanced distribution, posing a challenge for machine learning models to effectively handle imbalanced class scenarios. In the Adult dataset, we consider the race groups "white," "black," and "other" as protected groups. We assign the value of $\epsilon$ as 0.05 and set the noise level to 0.3 during training across all the algorithms.

**Evaluation metrics.** We present the Top-1 accuracies for the training and testing segments of the CIFAR10-ST and CIFAR-100-ST datasets (please see Figures 1 and 2 in Section 6). Furthermore, in addition to training and testing performance, we also include the maximum violation values for both the training and testing sections of the Adult dataset. Specifically, the maximum group violation is evaluated following Haddadpour et al. (2022). To ensure equal opportunities among different groups, even when group membership is uncertain and fluctuating during training, the objective is to develop a solution that is robust across various protected groups in the problem. We assume that we have access to the probability distribution of the actual group memberships ($P(gi = j|g^i = k)$ where $g^i$ represents the true group membership and $g^i$ represents the noisy group membership). With this information, we aim to enforce fairness constraints by considering all potential proxy groups based on this probability distribution, which can significantly increase the number of constraints. In the case of equal opportunity, our goal is to ensure that the true positive rate ($TPR$) for each group closely aligns with the $TPR$ of the overall dataset, within a certain threshold $\epsilon$. In other words, we want to achieve $tpr(g = j) \geq tpr(ALL) - \epsilon$ for every proxy group we define.

**Discussion.** In Figure 4, we evaluate the training performance on the adult dataset under the same conditions as mentioned earlier for testing in Section 6. Similar to the previous findings, in the leftmost image, we observe that FedDRO  outperforms both the constrained version of GCIVR and unconstrained baseline formulation. Evaluating the maximum group violation, we see the unconstrained optimization demonstrates the poorest performance, while our technique performs comparably to GCIVR, and improves in performance as the communication rounds increase. The right-most plot, confirms that increasing the local updates, i.e., $I$ results in improved performance, aligning with the theoretical guarantees presented in the paper.

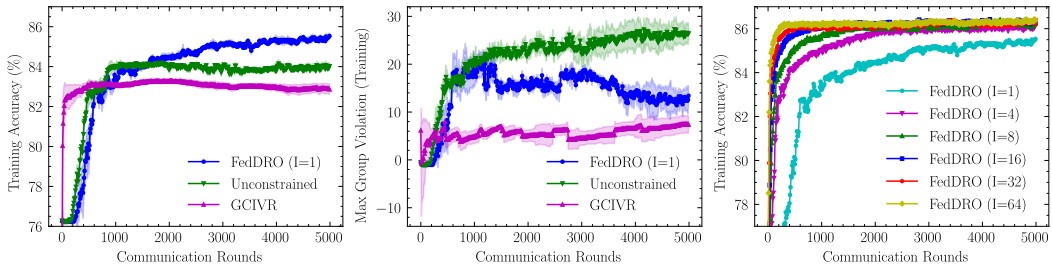

Figure 4: Comparison of training accuracies of FedDRO, GCIVR, and the unconstrained baseline (first two figures). Training performance of FedDRO with different $I$ (rightmost figure).

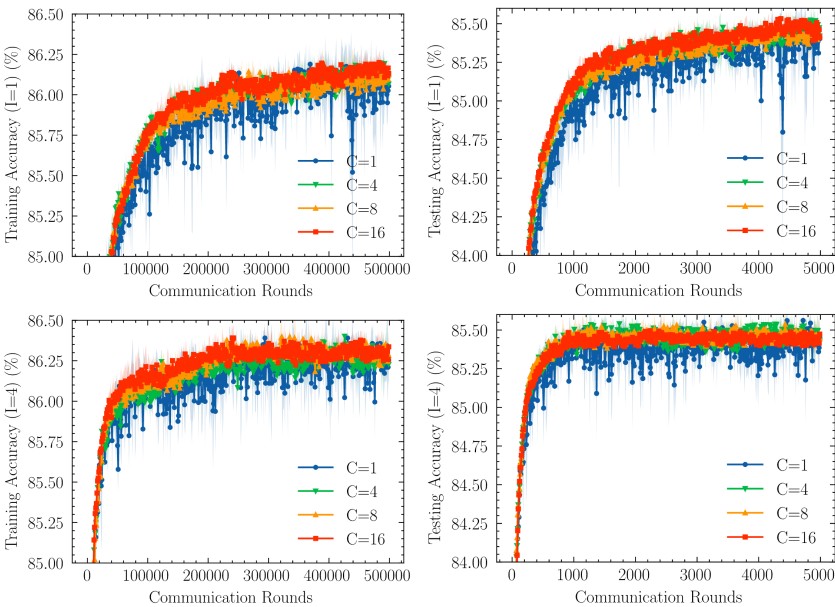

Figure 5: Training and testing performance of FedDRO with the number of clients (denoted as $C = 1, 2, 3$ and $4$ in the figure) and number of local updates, $I = 1$ and $4$.

In Figure 5, we evaluate the performance of FedDRO with the number of clients. Specifically, the accuracy demonstrates an upward trend as the value of $C$ (representing the number of clients) increases in the experiments conducted on the adult dataset. The top two plots depict the training and testing performance for $I = 1$, while the bottom two demonstrate the training and testing performance with $I = 4$.

## C USEFUL LEMMAS

**Lemma C.1.** *For vectors $a_1, a_2, \ldots, a_n \in \mathbb{R}^d$, we have*

$$\|a_1 + a_2 + \ldots, +a_n\|^2 \leq n\big[\|a_1\|^2 + \|a_2\|^2 + \ldots, +\|a_n\|^2\big].$$

**Lemma C.2.** *For a sequence of vectors $a_1, a_2, \ldots, a_K \in \mathbb{R}^d$, defining $\bar{a} \coloneqq \frac{1}{K}\sum_{k=1}^{K} a_k$, we then have*

$$\sum_{k=1}^{K} \|a_k - \bar{a}\|^2 \leq \sum_{k=1}^{K} \|a_k\|^2.$$

# D    PROOF OF THEOREM 4.1

We restate Theorem 4.1 for convenience.

**Theorem D.1** (Vanilla FedAvg: Non-Convergence for CO). *There exist functions $f(\cdot)$ and $g_k(\cdot)$ for $k \in [K]$ satisfying Assumptions 3.2, 3.3, and 3.4, and an initialization strategy such that for a fixed number of local updates $I > 1$, and for any $0 < \eta^t < C_\eta$ for $t \in \{0, 1, \ldots, T - 1\}$ where $C_\eta > 0$ is a constant, the iterates generated by Algorithm 1 under both Cases I and II do not converge to the stationary point of $\Phi(\cdot)$, where $\Phi(\cdot)$ is defined in (2) with $h(x) = 0$.*

*Proof.* We consider a setting where we have $K = 2$ nodes in the network. Also, let us consider a single-dimensional setting where the local functions $g_k : \mathbb{R} \to \mathbb{R}$ for $k = \{1, 2\}$ at each node are

$$g_1(x) := 4x - 4 \quad \text{and} \quad g_2(x) := -2x + 4.$$

Moreover, assume $f : \mathbb{R} \to \mathbb{R}$ as $f(y) := \sqrt{y^2 + 4}$. Therefore, the CO problem becomes

$$\min_{x \in \mathbb{R}} \left\{ \Phi(x) := f\left( \frac{1}{2}\Big(g_1(x) + g_2(x)\Big) \right) := \sqrt{\left[ \frac{1}{2}\Big(g_1(x) + g_2(x)\Big) \right]^2 + 4} = \sqrt{x^2 + 4} \right\}. \quad (10)$$

First, we establish that the functions $f(\cdot)$ and $g_k(\cdot)$ for $k \in [K]$ satisfy Assumptions 3.2, 3.3, and 3.4.

**Claim:** Functions $f$, $g_1$ and $g_2$ satisfy Assumptions 3.2, 3.3, and 3.4.

The above claim is straightforward to verify. Specifically, we have

- The functions $f$, $g_1$ and $g_2$ are differentiable and Lipschitz smooth.

- The function $f(\cdot)$ is Lipschitz. Moreover, $g_k(\cdot)$'s are deterministic functions implying mean-squared Lipschitzness.

- Assumption 3.3 is automatically satisfied since $g_k(\cdot)$'s are deterministic functions.

- Bounded heterogeneity of $g_k(\cdot)$'s is satisfied.

Note that it is clear from (10) that the minimizer of $\Phi(\cdot)$ is $x^* = 0$. In the following, we will show that Algorithm 1 is not suitable to solve such problems by establishing that there exists an initialization strategy and choice of step-sizes in the range $0 < \eta < C_\eta$ where $C_\eta > 0$ is a constant, the iterates generated by Algorithm 1 under both Cases I and II fail to converge to $x^*$. Next, we prove the statement of the theorem in two parts. In the first part, we tackle Case I of Algorithm 1 while in the second part, we prove Case II of Algorithm 1. Next, we consider Case I.

**Case I:** Let us first compute the local gradients at each agent. We have

$$\nabla \Phi_1(x) = \nabla g_1(x) \nabla f(y_1) = 4 \frac{y_1}{\sqrt{y_1^2 + 4}}$$
$$\nabla \Phi_2(x) = \nabla g_2(x) \nabla f(y_2) = -2 \frac{y_2}{\sqrt{y_2^2 + 4}}$$

To prove the results, we consider a simple setting with $I = 2$, i.e., each node conducts 2 local updates and shares the model parameters with the server. Moreover, we initialize the local iterates to be $x_k^0 = \bar{x}^0 = 0.5$ for $k = \{1, 2\}$ at both nodes. For this setting, let us write the update rule for Algorithm 1 in Case I.

1. Note that for every $t$ such that $t \bmod 2 = 0$, the local update at each node will be:

$$x_1^{t+1} = \bar{x}^t - 4\eta \frac{4\bar{x}^t - 4}{\sqrt{(4\bar{x}^t - 4)^2 + 4}}$$
$$x_2^{t+1} = \bar{x}^t + 2\eta \frac{-2\bar{x}^t + 4}{\sqrt{(-2\bar{x}^t + 4)^2 + 4}},$$

2. Moreover, the next immediate update at each node will be

$$x_1^{t+2} = x_1^{t+1} - 4\eta \frac{4x_1^{t+1} - 4}{\sqrt{(4x_1^{t+1} - 4)^2 + 4}}$$

$$x_2^{t+2} = x_2^{t+1} + 2\eta \frac{-2x_2^{t+1} + 4}{\sqrt{(-2x_2^{t+1} + 4)^2 + 4}},$$

3. This process keeps repeating for $T$ iterations.

Let us focus on the local functions $f(g_1(x))$ and $f(g_2(x))$. Note from the definition of $g_1(\cdot)$, $g_2(\cdot)$ and $f(\cdot)$ that the local optimum of these functions will be $x_1^* = 1$ and $x_2^* = 2$, respectively. Consequently, for appropriately chosen step-size $\eta$ in each iteration $x_1^{t+1}$ and $x_1^{t+2}$ at node 1 will converge towards $x_1^* = 1$ and similarly, $x_2^{t+1}$ and $x_2^{t+2}$ at node 2 will converge towards $x_1^* = 2$. This implies that we can expect the sequence $\bar{x}^t$ for each $t \in [T]$ to not converge to $x^* = 0$, the minimizer of the CO problem defined in (10). Let us present this argument formally.

**Claim:** For $C_\eta = 1/8$ such that we have $0 < \eta < C_\eta$, and utilizing the initialization $\bar{x}^0 = 0.5$, we have $\bar{x}^t \geq 0.5$ for every $t > 0$ with $t \bmod 2 = 0$.

This above Claim directly proves the statement of Theorem 4.1 for Case I. Let us now prove the claim formally. We utilize induction to prove the claim.

*Proof of claim:* First, note that the claim is automatically satisfied for $t = 0$ as a consequence of the initialization strategy. Assuming the claim holds for some $t \in [T]$ with $t \bmod 2 = 0$, i.e., we have $\bar{x}_t \geq 0.5$ for some $t \in [T]$ with $t \bmod 2 = 0$, we need to show that $\bar{x}_{t+2} \geq 0.5$.

In the following, we consider the following three cases: (1) $0.5 \leq \bar{x}_t < 1$, (2) $1 \leq \bar{x}_t < 2$, and (3) $\bar{x}_t \geq 2$. Here, we present the proof for case (1), the rest of the cases follow in a similar manner.

- Note from Step 1 above that since $0.5 \leq \bar{x}^t < 1$, we have $4\bar{x}^t - 4 < 0$ and $-2\bar{x}^t + 4 > 0$, which further implies that the locally updated iterates $x_1^{t+1} > \bar{x}^t \geq 0.5$ and $x_2^{t+1} > \bar{x}^t \geq 0.5$. Next, let us analyze the iterates at $t + 2$.

- At node 1, we further consider two cases, when $x_1^{t+1} < 1$ and the other when $x_1^{t+1} \geq 1$.

  - First, note that if $x_1^{t+1} < 1$ we will have $4x_1^{t+1} - 4 < 0$ in Step 2 above implying $x_1^{t+2} > x_1^{t+1} > \bar{x}^t \geq 0.5$.
  - Otherwise, if $x_1^{t+1} \geq 1$, we have $4x_1^{t+1} - 4 \geq 0$ however in this case we have

  $$\left| 4\eta \frac{4x_1^{t+1} - 4}{\sqrt{(4x_1^{t+1} - 4)^2 + 4}} \right| \leq 1/2 \text{ for } \eta \leq \frac{1}{8},$$

  again implying from the update rule in Step 2 that

  $$x_1^{t+2} \geq x_1^{t+1} - \frac{1}{2} \geq 0.5,$$

  where the last step follows from the fact that $x_1^{t+1} \geq 1$. Therefore, we have established that $x_1^{t+2} \geq 0.5$.

- At node 2, it is easy to establish that for case (1) with $0.5 \leq \bar{x}_t < 1$, we will have $0.5 \leq x_2^{t+1} \leq 1.5$. Note from the update rule in Step 2 that for this $x_2^{t+1}$, we have $-2x_2^{t+1} + 4 > 0$ which further implies that $x_2^{t+2} > x_2^{t+1} \geq 0.5$.

- Finally, we have established that both $x_1^{t+2} \geq 0.5$ and $x_2^{t+2} \geq 0.5$, implying $\bar{x}_{t+2} \geq 0.5$. This completes the proof of Case (1). Note that the proof for the other cases follows in a very similar straightforward manner.

Therefore, we have the proof of Case I in Algorithm 1. Next, we consider Case II where in addition to the model parameters, the local embeddings $g_k(\cdot)$ for $k \in [K]$ are also shared intermittently among nodes. Please see Case II in Algorithm 1.

**Case II:** Let us consider the same setting as in Case I. Specifically, we consider a simple setting with $I = 2$, i.e., each node conducts 2 local updates and shares the model parameters with the server. Moreover, we initialize the model parameters $x_k^0 = \bar{x}^0 = 0.5$ for $k = \{1, 2\}$ at both nodes. Note that this implies from the definition of $g_1(\cdot)$ and $g_2(\cdot)$ that $y_k^0 = \bar{y}^0 = 0.5$ for $k = \{1, 2\}$. For this setting, let us write the update rule for Algorithm 1.

1. Note that for every $t$ such that $t \bmod 2 = 0$, the local update at each node will be:

$$
x_1^{t+1} = \bar{x}^t - 4\eta \frac{\bar{x}^t}{\sqrt{(\bar{x}^t)^2 + 4}}
$$
$$
x_2^{t+1} = \bar{x}^t + 2\eta \frac{\bar{x}^t}{\sqrt{(\bar{x}^t)^2 + 4}},
$$

2. Moreover, the next immediate update at each node will be

$$
x_1^{t+2} = x_1^{t+1} - 4\eta \frac{4x_1^{t+1} - 4}{\sqrt{(4x_1^{t+1} - 4)^2 + 4}}
$$
$$
x_2^{t+2} = x_2^{t+1} + 2\eta \frac{-2x_2^{t+1} + 4}{\sqrt{(-2x_2^{t+1} + 4)^2 + 4}},
$$

3. This process keeps repeating for $T$ iterations.

We point out that this setting is considerably challenging compared to Case I since a cursory look at the algorithm may suggest that sharing the embeddings $g_k(\cdot)$ for $k \in [K]$ intermittently may help mitigate the bias in the gradient estimates. However, this is not the case as we show next.

**Claim:** For $C_\eta = 1/22$ such that we have $0 < \eta < C_\eta$, and utilizing the initialization $\bar{x}^0 = 0.5$, we have $\bar{x}^t \geq 0.5$ for every $t > 0$ with $t \bmod 2 = 0$.

We note that for this case the intuition is not as straightforward as in the previous case. We again prove the claim by induction.

*Proof of claim:* First, note that the claim is automatically satisfied for $t = 0$ as a consequence of the initialization strategy. Assuming the claim holds for some $t \in [T]$ with $t \bmod 2 = 0$, i.e., we have $\bar{x}_t \geq 0.5$ for some $t \in [T]$ with $t \bmod 2 = 0$, we need to show that $\bar{x}_{t+2} \geq 0.5$.

Let us first construct $x_1^{t+2}$ and $x_2^{t+2}$ as a function of $\bar{x}^t$. To this end, we have from the update rule in Steps 1 and 2 that

$$
x_1^{t+2} = \bar{x}^t(1 - \epsilon_1^t) - 4\eta \frac{4\bar{x}^t(1 - \epsilon_1^t) - 4}{\sqrt{(4\bar{x}^t(1 - \epsilon_1^t) - 4)^2 + 4}}
$$
$$
x_2^{t+2} = \bar{x}^t(1 + \epsilon_2^t) + 2\eta \frac{-2\bar{x}^t(1 + \epsilon_2^t) + 4}{\sqrt{(-2\bar{x}^t(1 + \epsilon_2^t) + 4)^2 + 4}},
$$

where we have defined $\epsilon_1^t := \frac{4\eta}{\sqrt{(\bar{x}^t)^2+4}}$ and $\epsilon_2^t := \frac{2\eta}{\sqrt{(\bar{x}^t)^2+4}}$, therefore, we have $\epsilon_1^t = 2\epsilon_2^t$. Using the above we can evaluate $\bar{x}^{t+2}$ as

$$
\begin{aligned}
\bar{x}^{t+2} &= \frac{1}{2}\left(x_1^{t+2} + x_2^{t+2}\right) \\
&= \left(\frac{2 - \epsilon_1^t + \epsilon_2^t}{2}\right)\bar{x}^t + 2\eta\frac{4 - 4\bar{x}^t\left(1 - \epsilon_1^t\right)}{\sqrt{(4\bar{x}^t\left(1 - \epsilon_1^t\right) - 4)^2 + 4}} + \eta\frac{4 - 2\bar{x}^t\left(1 + \epsilon_2^t\right)}{\sqrt{(-2\bar{x}^t\left(1 + \epsilon_2^t\right) + 4)^2 + 4}} \\
&= \left(1 - \frac{\epsilon_2^t}{2}\right)\bar{x}^t + 2\eta\frac{4 - 4\bar{x}^t\left(1 - \epsilon_1^t\right)}{\sqrt{(4\bar{x}^t\left(1 - \epsilon_1^t\right) - 4)^2 + 4}} + \eta\frac{4 - 2\bar{x}^t\left(1 + \epsilon_2^t\right)}{\sqrt{(-2\bar{x}^t\left(1 + \epsilon_2^t\right) + 4)^2 + 4}},
\end{aligned}
$$

where in the first term of the last equality, we have used the fact that $\epsilon_1^t = 2\epsilon_2^t$. Recall from the induction hypothesis that we have $\bar{x}^t \geq 0.5$, and we need to show that $\bar{x}^{t+2} \geq 0.5$. Note from above that to establish $\bar{x}^{t+2} \geq 0.5$, it suffices to show that

$$
\bar{x}^t - 0.5 + 2\eta\frac{4 - 4\bar{x}^t\left(1 - \epsilon_1^t\right)}{\sqrt{(4\bar{x}^t\left(1 - \epsilon_1^t\right) - 4)^2 + 4}} + \eta\frac{4 - 2\bar{x}^t\left(1 + \epsilon_2^t\right)}{\sqrt{(-2\bar{x}^t\left(1 + \epsilon_2^t\right) + 4)^2 + 4}} \geq \frac{\epsilon_2^t}{2}\bar{x}^t. \tag{11}
$$

From the definition of $\epsilon_2^t := \frac{2\eta}{\sqrt{(\bar{x}^t)^2+4}}$, we note that the r.h.s. term can be further upper bounded as

$$
\frac{\epsilon_2^t}{2}\bar{x}^t = \eta\frac{\bar{x}^t}{\sqrt{(\bar{x}^t)^2 + 4}} \leq \eta.
$$

Therefore, to establish to establish $\bar{x}^{t+2} \geq 0.5$, it suffices to show that

$$
\bar{x}^t - 0.5 + 2\eta\frac{4 - 4\bar{x}^t\left(1 - 2\epsilon_2^t\right)}{\sqrt{(4\bar{x}^t\left(1 - 2\epsilon_2^t\right) - 4)^2 + 4}} + \eta\frac{4 - 2\bar{x}^t\left(1 + \epsilon_2^t\right)}{\sqrt{(-2\bar{x}^t\left(1 + \epsilon_2^t\right) + 4)^2 + 4}} \geq \eta, \tag{12}
$$

where we have replaced $\epsilon_1^t = 2\epsilon_2^t$. Similar to the previous proof here we again consider three cases as listed below

- Case (1): $\frac{4-4\bar{x}^t(1-2\epsilon_2^t)}{\sqrt{(4\bar{x}^t(1-2\epsilon_2^t)-4)^2+4}} < 0$ and $\frac{4-2\bar{x}^t(1+\epsilon_2^t)}{\sqrt{(-2\bar{x}^t(1+\epsilon_2^t)+4)^2+4}} < 0$

- Case (2): $\frac{4-4\bar{x}^t(1-2\epsilon_2^t)}{\sqrt{(4\bar{x}^t(1-2\epsilon_2^t)-4)^2+4}} < 0$ and $\frac{4-2\bar{x}^t(1+\epsilon_2^t)}{\sqrt{(-2\bar{x}^t(1+\epsilon_2^t)+4)^2+4}} > 0$

- Case (3): $\frac{4-4\bar{x}^t(1-2\epsilon_2^t)}{\sqrt{(4\bar{x}^t(1-2\epsilon_2^t)-4)^2+4}} \geq 0$ and $\frac{4-2\bar{x}^t(1+\epsilon_2^t)}{\sqrt{(-2\bar{x}^t(1+\epsilon_2^t)+4)^2+4}} \geq 0$

We first consider Case (1). Note that Case (1) implies that $\bar{x}^t > 1$, and using the fact that $\frac{4-4\bar{x}^t(1-2\epsilon_2^t)}{\sqrt{(4\bar{x}^t(1-2\epsilon_2^t)-4)^2+4}} \geq -1$ and $\frac{4-2\bar{x}^t(1+\epsilon_2^t)}{\sqrt{(-2\bar{x}^t(1+\epsilon_2^t)+4)^2+4}} \geq -1$, we get

$$
\bar{x}^t - 0.5 + 2\eta\frac{4 - 4\bar{x}^t\left(1 - 2\epsilon_2^t\right)}{\sqrt{(4\bar{x}^t\left(1 - 2\epsilon_2^t\right) - 4)^2 + 4}} + \eta\frac{4 - 2\bar{x}^t\left(1 + \epsilon_2^t\right)}{\sqrt{(-2\bar{x}^t\left(1 + \epsilon_2^t\right) + 4)^2 + 4}} \geq 0.5 - 3\eta
$$

Note that by choosing $\eta \leq 1/8$, the sufficient condition in (12) is satisfied, which further implies that under Case (1), we have $\bar{x}^{t+2} \geq 0.5$. Next, we consider Case (2).

Note that for Case (2) we have $2/(1 + \epsilon_2^t) > \bar{x}^t > 1$, next using the fact that $\frac{4-4\bar{x}^t(1-2\epsilon_2^t)}{\sqrt{(4\bar{x}^t(1-2\epsilon_2^t)-4)^2+4}} \geq -1$ and $\frac{4-2\bar{x}^t(1+\epsilon_2^t)}{\sqrt{(-2\bar{x}^t(1+\epsilon_2^t)+4)^2+4}} \geq 0$, we get

$$
\bar{x}^t - 0.5 + 2\eta\frac{4 - 4\bar{x}^t\left(1 - 2\epsilon_2^t\right)}{\sqrt{(4\bar{x}^t\left(1 - 2\epsilon_2^t\right) - 4)^2 + 4}} + \eta\frac{4 - 2\bar{x}^t\left(1 + \epsilon_2^t\right)}{\sqrt{(-2\bar{x}^t\left(1 + \epsilon_2^t\right) + 4)^2 + 4}} \geq 0.5 - 2\eta
$$

Again choosing $\eta \leq 1/8$, the sufficient condition in (12) is satisfied, which further implies that under Case (2), we have $\bar{x}^{t+2} \geq 0.5$.

Finally, we consider the most challenging Case (3). Note that in Case (3) we have $0.5 \leq \bar{x}^t \leq 1/(1 - 2\epsilon_2^t)$. For this case, we revisit the sufficient condition in (11) and make it tight. Recall that we had from (11) that

$$\bar{x}^t - 0.5 + 2\eta \frac{4 - 4\bar{x}^t(1 - 2\epsilon_2^t)}{\sqrt{(4\bar{x}^t(1 - 2\epsilon_2^t) - 4)^2 + 4}} + \eta \frac{4 - 2\bar{x}^t(1 + \epsilon_2^t)}{\sqrt{(-2\bar{x}^t(1 + \epsilon_2^t) + 4)^2 + 4}} \geq \eta \frac{\bar{x}_t}{\sqrt{(\bar{x}_t)^2 + 4}},$$

now using the fact that for Case (3), we have $0.5 \leq \bar{x}^t \leq 1/(1 - 2\epsilon_2^t)$, we can restate the sufficient condition as

$$\bar{x}^t - 0.5 + 2\eta \frac{4 - 4\bar{x}^t(1 - 2\epsilon_2^t)}{\sqrt{(4\bar{x}^t(1 - 2\epsilon_2^t) - 4)^2 + 4}} + \eta \frac{4 - 2\bar{x}^t(1 + \epsilon_2^t)}{\sqrt{(-2\bar{x}^t(1 + \epsilon_2^t) + 4)^2 + 4}} \geq \frac{\eta}{2}, \quad (13)$$

where we have used the fact that $0.5 \leq \bar{x}^t \leq 1.1$ for $\eta < 1/22$ and the fact that the term $\eta \frac{\bar{x}_t}{\sqrt{(\bar{x}_t)^2 + 4}} > \frac{\eta}{2}$ for $0.5 \leq \bar{x}^t \leq 1.1$. Moreover, $\eta < 1/22$ ensures that $1 + \epsilon_2^t \leq 23/22$. Next, using the fact that $\frac{4 - 4\bar{x}^t(1 - 2\epsilon_2^t)}{\sqrt{(4\bar{x}^t(1 - 2\epsilon_2^t) - 4)^2 + 4}} > 0$ and

$$\frac{4 - 2\bar{x}^t(1 + \epsilon_2^t)}{\sqrt{(-2\bar{x}^t(1 + \epsilon_2^t) + 4)^2 + 4}} \geq \frac{4 - 2\bar{x}^t(23/22)}{\sqrt{(-2\bar{x}^t(1 + \epsilon_2^t) + 4)^2 + 4}} \geq \frac{6}{10},$$

Substituting in the l.h.s. of the sufficient condition stated in (13), we get

$$\bar{x}^t - 0.5 + 2\eta \frac{4 - 4\bar{x}^t(1 - 2\epsilon_2^t)}{\sqrt{(4\bar{x}^t(1 - 2\epsilon_2^t) - 4)^2 + 4}} + \eta \frac{4 - 2\bar{x}^t(1 + \epsilon_2^t)}{\sqrt{(-2\bar{x}^t(1 + \epsilon_2^t) + 4)^2 + 4}} \geq \frac{6\eta}{10},$$

where we used that fact that $\bar{x}^t \geq 0.5$. Note that $\frac{6\eta}{10} > \frac{\eta}{2}$, therefore, the sufficient condition stated in (13) is satisfied. This further implies that the $\bar{x}^{t+2} \geq 0.5$ during the execution of the algorithm.

Recall that the optimal solution for solving the CO problem is $x^* = 0$. This means Algorithm 1 under both Case I and II fails to converge to the stationary solution.

Hence, the theorem is proved. $\qquad \square$

Finally, we corroborate the result presented in Theorem D.1 via numerical experiment for solving (10) using Case II of Algorithm 1. In Figure 6, we plot the evolution of $\bar{x}^t$ in each communication round. We note that $\bar{x}^t$ is lower bounded by $0.5$ as established in the proof of Theorem 4.2 above. In fact, note that for all the settings as the communication rounds increase, $\bar{x}^t$ eventually converges to a quantity that is greater than 1. However, as discussed for the example considered to establish the proof of Theorem 4.1, we know that the true optimizer of the CO problem (10) is $x^* = 0$.

## E    PROOF OF THEOREM 4.2

**Theorem E.1** (Modified FedAvg: Convergence for CO). *Suppose we modify Algorithm 1 such that $y_k^t = \bar{y}^t$ is updated at each iteration $t \in \{0, 1, \ldots, T - 1\}$ instead of $[t + 1 \mod I]$ iterations as in current version of Algorithm 1. Then if functions $f(\cdot)$ and $g_k(x)$ for $k \in [K]$ satisfy Assumptions 3.2, 3.3, and 3.4 such that for a fixed number of local updates $1 \leq I \leq \mathcal{O}(T^{1/4})$, there exists a choice of $\eta^t > 0$ for $t \in \{0, 1, \ldots, T - 1\}$ such that the iterates generated by (modified) Algorithm 1 converge to the stationary point of $\Phi(\cdot)$, where $\Phi(\cdot)$ is defined in (2) with $h(x) = 0$.*

*Proof.* Theorem E.1 is a direct consequence of Theorem 5.1. Therefore, we next prove the main result of the paper in Theorem 5.1. $\qquad \square$

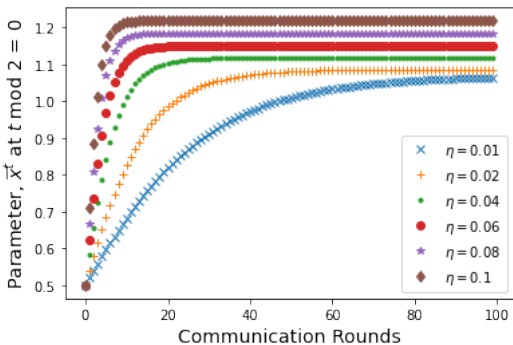

Figure 6: The evolution of parameter $\bar{x}^t$ at each communication round for different choices of step-sizes $\eta$.

## F   PROOF OF THEOREM 5.1

For the purpose of this proof, we define the filtration $\mathcal{F}^t$ as the sigma-algebra generated by the iterates $x_k^1, x_k^1, \ldots, x_k^t$ as

$$\mathcal{F}^t \coloneqq \sigma(x_k^1, x_k^1, \ldots, x_k^t, \text{ for all } k \in [K]).$$

Moreover, we define the following. Assuming the total training rounds, $T - 1$, to be a multiple of $I$, i.e., $T - 1 = S \times I$ for some $S \in \mathbb{N}$, we define $t_s \coloneqq s \times I$ with $s \in \{0, 1, \ldots, S\}$ as the training rounds where the potentially high-dimensional model parameters, $x_k^t$, are shared among the clients. Next, we state Theorem 5.1 again and present the detailed proof of the result.

**Theorem F.1.** *Under Assumptions 3.2, 3.3, and 3.4 and with the choice of step-size $\eta^t = \eta = \sqrt{\frac{|b|K}{T}}$ for all $t \in \{0, 1, \ldots, T-1\}$. Moreover, choosing the momentum parameter $\beta^t = \beta = c_\beta \eta$ where $c_\beta = 4B_g^4 L_f^2$. Then for*

$$T \geq T_{th} \coloneqq \max\left\{ \frac{4(L_\Phi|b|K + 8B_g^2)^2}{|b|K}, \frac{B_g^4(96L_h^2 + 96B_f^2 L_g^2)^2}{|b|K(L_h^2 + 2B_f^2 L_g^2 + 4B_g^4 L_f^2)^2}, \right.$$
$$\left. (216L_h^2 + 216B_f^2 L_g^2)I^2|b|K \right\}$$

*The iterates generated by Algorithm 2 satisfy*

$$\mathbb{E}\big\|\nabla\Phi(\bar{x}^{a(T)})\big\|^2 \leq \frac{2\big[\Phi(\bar{x}^0) - \Phi(x^*) + \big\|\bar{y}^0 - g(\bar{x}^0)\big\|^2\big]}{\sqrt{|b|KT}} + \frac{K(I-1)^2}{T}\big[2\bar{L}_{f,g}\sigma_h^2 + 2B_f^2\bar{L}_{f,g}\sigma_g^2\big]$$
$$+ \frac{1}{\sqrt{|b|KT}}\big[(4L_\Phi + 8B_g^2)\sigma_h^2 + (4L_\Phi B_f^2 + 4c_\beta^2 + 8B_f^2 B_g^2)\sigma_g^2\big]$$
$$+ \frac{|b|K(I-1)^2}{T}\big[6\bar{L}_{f,g}\Delta_h^2 + 6B_f^2\bar{L}_{f,g}\Delta_g^2\big] + \frac{1}{\sqrt{|b|KT}}\big[96B_g^2\,\Delta_h^2 + 96B_f^2 B_g^2\,\Delta_g^2\big].$$

**Corollary F.2.** *Under the same setting as Theorem 5.1, for the choice of local updates $I = T^{1/4}/(|b|K)^{3/4}$, the iterates generated by Algorithm 2 satisfy*

$$\mathbb{E}\big\|\nabla\Phi(\bar{x}^{a(T)})\big\|^2 \leq \frac{2\big[\Phi(\bar{x}^0) - \Phi(x^*) + \big\|\bar{y}^0 - g(\bar{x}^0)\big\|^2\big]}{\sqrt{|b|KT}} + \frac{C_{\sigma_h}}{\sqrt{|b|KT}}\sigma_h^2 + \frac{C_{\sigma_g}}{\sqrt{|b|KT}}\sigma_g^2$$
$$+ \frac{C_{\Delta_h}}{\sqrt{|b|KT}}\Delta_h^2 + \frac{C_{\Delta_g}}{\sqrt{|b|KT}}\Delta_g^2. \tag{14}$$

*where the constants $C_{\sigma_h}$, $C_{\sigma_g}$, $C_{\Delta_h}$, and $C_{\Delta_g}$ are constants dependent on $L_g$, $L_h$, $L_f$, $B_g$, and $B_f$.*

We prove the Theorem in multiple steps with the help of several intermediate Lemmas.

**Lemma F.3 (Descent in Function Value).** *Under Assumptions 3.2-3.4, the iterates generated by Algorithm 2 satisfy*

$$\mathbb{E}\big[\Phi(\bar{x}^{t+1}) - \Phi(\bar{x}^t)\big] \leq -\frac{\eta^t}{2}\mathbb{E}\big\|\nabla\Phi(\bar{x}^t)\big\|^2 - \Big(\frac{\eta^t}{2} - (\eta^t)^2 L_\Phi\Big)\mathbb{E}\Big\|\frac{1}{K}\sum_{k=1}^{K}\mathbb{E}\big[\nabla\Phi_k(x_k^t;\bar{\xi}_k^t)\big|\mathcal{F}^t\big]\Big\|^2$$

$$+ \eta^t\big(L_h^2 + 2B_f^2 L_g^2 + 4B_g^4 L_F^2\big)\frac{1}{K}\sum_{k=1}^{K}\mathbb{E}\|x_k^t - \bar{x}^t\|^2 + 4B_g^4 L_f^2\eta^t\,\mathbb{E}\Big\|\bar{y}^t - \frac{1}{K}\sum_{k=1}^{K}g_k(x_k^t)\Big\|^2$$

$$+ \frac{2(\eta^t)^2 L_\Phi}{K|b_h|}\sigma_h^2 + \frac{2(\eta^t)^2 L_\Phi B_f^2}{K|b_g|}\sigma_g^2.$$

*for all $t \in \{0, 1, \ldots, T-1\}$.*

*Proof.* Using the fact that the loss function $\Phi(x)$ is $L_\Phi$-Lipschitz smooth, we get

$$\mathbb{E}\big[\Phi(\bar{x}^{t+1}) - \Phi(\bar{x}^t)\big]$$

$$\leq \mathbb{E}\Big[\langle\nabla\Phi(\bar{x}^t), \bar{x}^{t+1} - \bar{x}^t\rangle + \frac{L_\Phi}{2}\|\bar{x}^{t+1} - \bar{x}^t\|^2\Big]$$

$$\stackrel{(a)}{\leq} \mathbb{E}\Big[-\eta^t\Big\langle\nabla\Phi(\bar{x}^t), \frac{1}{K}\sum_{k=1}^{K}\nabla\Phi_k(x_k^t;\bar{\xi}_k^t)\Big\rangle + \frac{(\eta^t)^2 L_\Phi}{2}\Big\|\frac{1}{K}\sum_{k=1}^{K}\nabla\Phi_k(x_k^t;\bar{\xi}_k^t)\Big\|^2\Big]$$

$$\stackrel{(b)}{\leq} \mathbb{E}\Big[-\eta^t\Big\langle\nabla\Phi(\bar{x}^t), \frac{1}{K}\sum_{k=1}^{K}\mathbb{E}\big[\nabla\Phi_k(x_k^t;\bar{\xi}_k^t)\big|\mathcal{F}^t\big]\Big\rangle + \frac{(\eta^t)^2 L_\Phi}{2}\Big\|\frac{1}{K}\sum_{k=1}^{K}\nabla\Phi_k(x_k^t;\bar{\xi}_k^t)\Big\|^2\Big]$$

$$\stackrel{(c)}{\leq} -\frac{\eta^t}{2}\mathbb{E}\big\|\nabla\Phi(\bar{x}^t)\big\|^2 - \Big(\frac{\eta^t}{2} - (\eta^t)^2 L_\Phi\Big)\mathbb{E}\Big\|\frac{1}{K}\sum_{k=1}^{K}\mathbb{E}\big[\nabla\Phi_k(x_k^t;\bar{\xi}_k^t)\big|\mathcal{F}^t\big]\Big\|^2$$

$$+ \frac{\eta^t}{2}\underbrace{\mathbb{E}\Big\|\nabla\Phi(\bar{x}^t) - \frac{1}{K}\sum_{k=1}^{K}\mathbb{E}\big[\nabla\Phi_k(x_k^t;\bar{\xi}_k^t)\big|\mathcal{F}^t\big]\Big\|^2}_{\text{Term I}} \tag{15}$$

$$+ (\eta^t)^2 L_\Phi\underbrace{\mathbb{E}\Big\|\frac{1}{K}\sum_{k=1}^{K}\nabla\Phi_k(x_k^t;\bar{\xi}_k^t) - \frac{1}{K}\sum_{k=1}^{K}\mathbb{E}\big[\nabla\Phi_k(x_k^t;\bar{\xi}_k^t)\big|\mathcal{F}^t\big]\Big\|^2}_{\text{Term II}},$$

where $(a)$ follows from the update step in Algorithm 2; $(b)$ results from moving the conditional expectation w.r.t. the filtration $\mathcal{F}^t$ inside the inner-product; finally, $(c)$ uses the equality $2\langle a, b\rangle = \|a\|^2 + \|b\|^2 - \|a - b\|^2$ for $a, b \in \mathbb{R}^d$ and Lemma C.1 to split the last term.

Next, we consider Terms I and II separately. First, note that from the definition of $\nabla\Phi_k(x_k^t;\bar{\xi}_k^t)$ for all $k \in [K]$, we have

$$\mathbb{E}\big[\nabla\Phi_k(x_k^t;\bar{\xi}_k^t)\big|\mathcal{F}^t\big] = \mathbb{E}\Big[\frac{1}{|b_{h_k}^t|}\sum_{i \in b_{h_k}^t}\nabla h_k(x_k^t;\xi_{k,i}^t) + \frac{1}{|b_{g_k}^t|}\sum_{j \in b_{g_k}^t}\nabla g_k(x_k^t;\zeta_{k,j}^t)\nabla f(\bar{y}^t)\Big|\mathcal{F}^t\Big]$$

$$\stackrel{(a)}{=} \nabla h_k(x_k^t) + \nabla g_k(x_k^t)\nabla f(\bar{y}^t) \tag{16}$$

where $(a)$ follows from Assumption 3.3. Moreover, from the definition of $\Phi(\bar{x}^t)$, we have

$$\nabla\Phi(\bar{x}^t) = \frac{1}{K}\sum_{k=1}^{K}\Big[\nabla h_k(\bar{x}^t) + \nabla g_k(\bar{x}^t)\nabla f(g(\bar{x}^t))\Big], \tag{17}$$

where $g(\bar{x}^t) = \frac{1}{K}\sum_{k=1}^K g_k(\bar{x}^t)$. Next, utilizing the expressions obtained in (16) and (17) we bound Term I as

$$\text{Term I} := \mathbb{E}\left\|\nabla\Phi(\bar{x}^t) - \frac{1}{K}\sum_{k=1}^K \mathbb{E}\big[\nabla\Phi_k(x_k^t;\bar{\xi}_k^t)\big|\mathcal{F}^t\big]\right\|^2$$

$$= \mathbb{E}\left\|\frac{1}{K}\sum_{k=1}^K \Big[\nabla h_k(\bar{x}^t) + \nabla g_k(\bar{x}^t)\nabla f(g(\bar{x}^t)) - \big[\nabla h_k(x_k^t) + \nabla g_k(x_k^t)\nabla f(\bar{y}^t)\big]\Big]\right\|^2$$

$$\overset{(a)}{\leq} \frac{2}{K}\sum_{k=1}^K \Big[\mathbb{E}\|\nabla h_k(x_k^t) - \nabla h_k(\bar{x}^t)\|^2 + \|\nabla g_k(x_k^t)\nabla f(\bar{y}^t) - \nabla g_k(\bar{x}^t)\nabla f(g(\bar{x}^t))\|^2\Big]$$

$$\overset{(b)}{\leq} \frac{2L_h^2}{K}\sum_{k=1}^K \mathbb{E}\|x_k^t - \bar{x}^t\|^2 + \frac{4}{K}\sum_{k=1}^K \mathbb{E}\big\|\nabla g_k(x_k^t)\big[\nabla f(\bar{y}^t) - \nabla f(g(\bar{x}^t))\big]\big\|^2$$

$$+ \frac{4}{K}\sum_{k=1}^K \mathbb{E}\big\|\big[\nabla g_k(x_k^t) - \nabla g_k(\bar{x}^t)\big]\nabla f(g(\bar{x}^t))\big\|^2$$

$$\overset{(c)}{\leq} \frac{2L_h^2}{K}\sum_{k=1}^K \mathbb{E}\|x_k^t - \bar{x}^t\|^2 + \frac{4B_g^2}{K}\sum_{k=1}^K \mathbb{E}\big\|\nabla f(\bar{y}^t) - \nabla f(g(\bar{x}^t))\big\|^2$$

$$+ \frac{4B_f^2}{K}\sum_{k=1}^K \mathbb{E}\|\nabla g_k(x_k^t) - \nabla g_k(\bar{x}^t)\|^2$$

$$\overset{(d)}{\leq} \left(\frac{2L_h^2}{K} + \frac{4B_f^2L_g^2}{K}\right)\sum_{k=1}^K \mathbb{E}\|x_k^t - \bar{x}^t\|^2 + 4B_g^2 L_f^2 \underbrace{\mathbb{E}\big\|\bar{y}^t - g(\bar{x}^t)\big\|^2}_{\text{Term III}}.$$

Next, let us consider Term III above.

$$\text{Term III} := \mathbb{E}\big\|\bar{y}^t - g(\bar{x}^t)\big\|^2$$

$$\overset{(a)}{\leq} 2\mathbb{E}\left\|\bar{y}^t - \frac{1}{K}\sum_{k=1}^K g_k(x_k^t)\right\|^2 + 2\mathbb{E}\left\|\frac{1}{K}\sum_{k=1}^K g_k(x_k^t) - g(\bar{x}^t)\right\|^2$$

$$\overset{(b)}{\leq} 2\mathbb{E}\left\|\bar{y}^t - \frac{1}{K}\sum_{k=1}^K g_k(x_k^t)\right\|^2 + \frac{2}{K}\sum_{k=1}^K \mathbb{E}\big\|g_k(x_k^t) - g_k(\bar{x}^t)\big\|^2$$

$$\overset{(c)}{\leq} 2\mathbb{E}\left\|\bar{y}^t - \frac{1}{K}\sum_{k=1}^K g_k(x_k^t)\right\|^2 + \frac{2B_g^2}{K}\sum_{k=1}^K \mathbb{E}\|x_k^t - \bar{x}^t\|^2,$$

where $(a)$ follows from the application of Lemma C.1; $(b)$ results from the definition of $g(x) = \frac{1}{K}\sum_{k=1}^K g_k(x)$ and the use of Lemma C.1; finally $(c)$ results from the Lipschitz-ness of $g_k(\cdot)$ for all $k \in [K]$.

Next, we consider Term II below

$$\text{Term II} := \mathbb{E}\left\|\frac{1}{K}\sum_{k=1}^K \nabla\Phi_k(x_k^t;\bar{\xi}_k^t) - \frac{1}{K}\sum_{k=1}^K \mathbb{E}\big[\nabla\Phi_k(x_k^t;\bar{\xi}_k^t)\big|\mathcal{F}^t\big]\right\|^2$$

$$\overset{(a)}{=} \frac{1}{K^2}\sum_{k=1}^K \mathbb{E}\big\|\nabla\Phi_k(x_k^t;\bar{\xi}_k^t) - \mathbb{E}\big[\nabla\Phi_k(x_k^t;\bar{\xi}_k^t)\big|\mathcal{F}^t\big]\big\|^2$$

$$\overset{(b)}{=} \frac{1}{K^2}\sum_{k=1}^K \mathbb{E}\left\|\frac{1}{|b_{h_k}^t|}\sum_{i\in b_{h_k}^t}\nabla h_k(x_k^t;\xi_{k,i}^t) + \frac{1}{|b_{g_k}^t|}\sum_{j\in b_{g_k}^t}\nabla g_k(x_k^t;\zeta_{k,j}^t)\nabla f(\bar{y}^t)\right.$$

$$\left. - \big[\nabla h_k(x_k^t) + \nabla g_k(x_k^t)\nabla f(\bar{y}^t)\big]\right\|^2$$

$$\stackrel{(c)}{=} \frac{2}{K^2} \sum_{k=1}^{K} \mathbb{E} \left\| \frac{1}{|b_{h_k}^t|} \sum_{i \in b_{h_k}^t} \nabla h_k(x_k^t; \xi_{k,i}^t) - \nabla h_k(x_k^t) \right\|^2$$

$$+ \frac{2}{K^2} \sum_{k=1}^{K} \mathbb{E} \left\| \frac{1}{|b_{g_k}^t|} \sum_{j \in b_{g_k}^t} \nabla g_k(x_k^t; \zeta_{k,j}^t) \nabla f(\bar{y}^t) - \nabla g_k(x_k^t) \nabla f(\bar{y}^t) \right\|^2$$

$$\stackrel{(d)}{\leq} \frac{2\sigma_h^2}{K|b_h|} + \frac{2\sigma_g^2 B_f^2}{K|b_g|},$$

where $(a)$ follows from the application of Lemma C.1; $(b)$ follows from the definition of the stochastic gradient in (7) and its expectation in (16); $(c)$ again uses Lemma C.1; Finally, $(d)$ uses Cauchy-Schwartz inequality, Lipschitzness of $f(\bar{y}^t)$ and Assumption 3.3 and using $|b_{h_k}| = |b_h|$ and $|b_{g_k}| = |b_g|$ for all $k \in [K]$.

Next, substituting the upper bounds obtained for Terms I, II, and III into (15), we get

$$\mathbb{E}\big[\Phi(\bar{x}^{t+1}) - \Phi(\bar{x}^t)\big] \leq -\frac{\eta^t}{2} \mathbb{E}\big\|\nabla\Phi(\bar{x}^t)\big\|^2 - \left(\frac{\eta^t}{2} - (\eta^t)^2 L_\Phi\right) \mathbb{E} \left\| \frac{1}{K} \sum_{k=1}^{K} \mathbb{E}\big[\nabla\Phi_k(x_k^t; \bar{\xi}_k^t)\big|\mathcal{F}^t\big] \right\|^2$$

$$+ \eta^t \big(L_h^2 + 2B_f^2 L_g^2 + 4B_g^4 L_F^2\big) \underbrace{\frac{1}{K} \sum_{k=1}^{K} \mathbb{E}\|x_k^t - \bar{x}^t\|^2}_{\text{Term IV}} + 4B_g^4 L_f^2 \eta^t \underbrace{\mathbb{E} \left\| \bar{y}^t - \frac{1}{K} \sum_{k=1}^{K} g_k(x_k^t) \right\|^2}_{\text{Term V}}$$

$$+ \frac{2(\eta^t)^2 L_\Phi}{K|b_h|} \sigma_h^2 + \frac{2(\eta^t)^2 L_\Phi B_f^2}{K|b_g|} \sigma_g^2. \tag{18}$$

Therefore, we have the proof of the Lemma. $\qquad\square$

Next, we bound Terms IV and V in (18) in the next Lemmas. Let us first consider Term IV.

**Lemma F.4 (Client Drift).** *Under Assumptions 3.2-3.4, the iterates generated by Algorithm 2 satisfy*

$$\frac{1}{K} \sum_{k=1}^{K} \mathbb{E}\|x_k^t - \bar{x}^t\| \leq (I-1)\Big(24L_h^2 + 24B_f^2 L_g^2\Big) \sum_{\ell=t_s}^{t-1} \frac{(\eta^\ell)^2}{K} \sum_{k=1}^{K} \mathbb{E}\|x_k^\ell - \bar{x}^\ell\|^2$$

$$+ (I-1)\left(\frac{4}{|b_h^t|}\sigma_h^2 + \frac{4B_f^2}{|b_g^t|}\sigma_g^2\right) \sum_{\ell=t_s}^{t-1} (\eta^\ell)^2 + (I-1)\Big(12\Delta_h^2 + 12B_f^2\Delta_g^2\Big) \sum_{\ell=t_s}^{t-1} (\eta^\ell)^2.$$

*Proof.* Recall from the definition of $t_s$ that we have $x_k^{t_s} = \bar{x}^{t_s}$ for all $s \in \{0, 1, \ldots, S\}$. Next, we have from the update rule in Algorithm 2 that for all $t \in [t_s + 1, t_{s+1} - 1]$

$$x_k^t = x_k^{t-1} - \eta^{t-1}\nabla\Phi_k(x_k^{t-1}; \bar{\xi}_k^{t-1}) \stackrel{(a)}{=} x_k^{t_s} - \sum_{\ell=t_s}^{t-1} \eta^\ell \nabla\Phi_k(x_k^\ell; \bar{\xi}_k^\ell). \tag{19}$$

where $(a)$ results from unrolling the updates from Algorithm 2. Similarly, we have

$$\bar{x}^t = \bar{x}^{t-1} - \eta^{t-1}\frac{1}{K}\sum_{k=1}^{K}\nabla\Phi_k(x_k^{t-1}; \bar{\xi}_k^{t-1}) = \bar{x}^{t_s} - \frac{1}{K}\sum_{k=1}^{K}\sum_{\ell=t_s}^{t-1} \eta^\ell \nabla\Phi_k(x_k^\ell; \bar{\xi}_k^\ell) \tag{20}$$

Bounding Term IV, we have

$$
\begin{aligned}
\text{Term IV} &:= \frac{1}{K} \sum_{k=1}^{K} \mathbb{E} \| x_k^t - \bar{x}^t \|^2 \\
&\overset{(a)}{=} \frac{1}{K} \sum_{k=1}^{K} \mathbb{E} \left\| \sum_{\ell=t_s}^{t-1} \eta^\ell \nabla \Phi_k(x_k^\ell; \bar{\xi}_k^\ell) - \frac{1}{K} \sum_{k=1}^{K} \sum_{\ell=t_s}^{t-1} \eta^\ell \nabla \Phi_k(x_k^\ell; \bar{\xi}_k^\ell) \right\|^2 \\
&\overset{(b)}{=} (I-1) \sum_{\ell=t_s}^{t-1} \frac{(\eta^\ell)^2}{K} \sum_{k=1}^{K} \underbrace{\mathbb{E} \left\| \nabla \Phi_k(x_k^\ell; \bar{\xi}_k^\ell) - \frac{1}{K} \sum_{k=1}^{K} \nabla \Phi_k(x_k^\ell; \bar{\xi}_k^\ell) \right\|^2}_{\text{Term VI}}
\end{aligned}
$$

where $(a)$ follows from (19) and (20) and $(b)$ follows from the application of Lemma C.1. Next, we bound Term VI in the above expression.

$$
\begin{aligned}
\text{Term VI} &:= \mathbb{E} \left\| \nabla \Phi_k(x_k^\ell; \bar{\xi}_k^\ell) - \frac{1}{K} \sum_{k=1}^{K} \nabla \Phi_k(x_k^\ell; \bar{\xi}_k^\ell) \right\|^2 \\
&\overset{(a)}{=} \mathbb{E} \left\| \frac{1}{|b_{h_k}^\ell|} \sum_{i \in b_{h_k}^\ell} \nabla h_k(x_k^\ell; \xi_{k,i}^\ell) + \frac{1}{|b_{g_k}^\ell|} \sum_{j \in b_{g_k}^\ell} \nabla g_k(x_k^\ell; \zeta_{k,j}^\ell) \nabla f(\bar{y}^\ell) \right. \\
&\quad \left. - \frac{1}{K} \sum_{k=1}^{K} \left[ \frac{1}{|b_{h_k}^\ell|} \sum_{i \in b_{h_k}^\ell} \nabla h_k(x_k^\ell; \xi_{k,i}^\ell) + \frac{1}{|b_{g_k}^\ell|} \sum_{j \in b_{g_k}^\ell} \nabla g_k(x_k^\ell; \zeta_{k,j}^\ell) \nabla f(\bar{y}^\ell) \right] \right\|^2 \\
&\overset{(b)}{\leq} 2\mathbb{E} \left\| \frac{1}{|b_{h_k}^\ell|} \sum_{i \in b_{h_k}^\ell} \nabla h_k(x_k^\ell; \xi_{k,i}^\ell) - \frac{1}{K} \sum_{k=1}^{K} \frac{1}{|b_{h_k}^\ell|} \sum_{i \in b_{h_k}^\ell} \nabla h_k(x_k^\ell; \xi_{k,i}^\ell) \right\|^2 \\
&\quad + 2\mathbb{E} \left\| \frac{1}{|b_{g_k}^\ell|} \sum_{j \in b_{g_k}^\ell} \nabla g_k(x_k^\ell; \zeta_{k,j}^\ell) \nabla f(\bar{y}^\ell) - \frac{1}{K} \sum_{k=1}^{K} \frac{1}{|b_{g_k}^\ell|} \sum_{j \in b_{g_k}^\ell} \nabla g_k(x_k^\ell; \zeta_{k,j}^t) \nabla f(\bar{y}^\ell) \right\|^2 \\
&\overset{(c)}{\leq} 2 \underbrace{\mathbb{E} \left\| \frac{1}{|b_{h_k}^\ell|} \sum_{i \in b_{h_k}^\ell} \nabla h_k(x_k^\ell; \xi_{k,i}^\ell) - \frac{1}{K} \sum_{k=1}^{K} \frac{1}{|b_{h_k}^\ell|} \sum_{i \in b_{h_k}^\ell} \nabla h_k(x_k^\ell; \xi_{k,i}^\ell) \right\|^2}_{\text{Term VII}} \\
&\quad + 2B_f^2 \underbrace{\mathbb{E} \left\| \frac{1}{|b_{g_k}^\ell|} \sum_{j \in b_{g_k}^\ell} \nabla g_k(x_k^\ell; \zeta_{k,j}^\ell) - \frac{1}{K} \sum_{k=1}^{K} \frac{1}{|b_{g_k}^\ell|} \sum_{j \in b_{g_k}^\ell} \nabla g_k(x_k^\ell; \zeta_{k,j}^\ell) \right\|^2}_{\text{Term VIII}},
\end{aligned}
$$

where $(a)$ results from the definition of the stochastic gradient evaluated in (7); $(b)$ uses Lemma C.1; and $(c)$ utilizes the Cauchy-Schwartz inequality combined with the Lipschitzness of $f(\cdot)$. Next, in order to upper bound Term VI, we bound Terms VII and VIII separately. First, let us consider

Term VII above

$$\text{Term VII} := \mathbb{E}\left\|\frac{1}{|b_{h_k}^\ell|}\sum_{i\in b_{h_k}^\ell}\nabla h_k(x_k^\ell;\xi_{k,i}^\ell) - \frac{1}{K}\sum_{k=1}^K\frac{1}{|b_{h_k}^\ell|}\sum_{i\in b_{h_k}^\ell}\nabla h_k(x_k^\ell;\xi_{k,i}^\ell)\right\|^2$$

$$\overset{(a)}{\le} 2\mathbb{E}\left\|\left[\frac{1}{|b_{h_k}^\ell|}\sum_{i\in b_{h_k}^\ell}\nabla h_k(x_k^\ell;\xi_{k,i}^\ell) - \nabla h_k(x_k^\ell)\right] - \frac{1}{K}\sum_{k=1}^K\left[\frac{1}{|b_{h_k}^\ell|}\sum_{i\in b_{h_k}^\ell}\nabla h_k(x_k^\ell;\xi_{k,i}^\ell) - \nabla h_k(x_k^\ell)\right]\right\|^2$$

$$+ 2\mathbb{E}\left\|\nabla h_k(x_k^\ell) - \frac{1}{K}\sum_{k=1}^K\nabla h_k(x_k^\ell)\right\|^2$$

$$\overset{(b)}{\le} 2\mathbb{E}\left\|\frac{1}{|b_{h_k}^\ell|}\sum_{i\in b_{h_k}^\ell}\nabla h_k(x_k^\ell;\xi_{k,i}^\ell) - \nabla h_k(x_k^\ell)\right\|^2 + 2\mathbb{E}\left\|\nabla h_k(x_k^\ell) - \frac{1}{K}\sum_{k=1}^K\nabla h_k(x_k^\ell)\right\|^2$$

$$\overset{(c)}{\le} \frac{2\sigma_h^2}{|b_{h_k}^\ell|} + \underbrace{2\,\mathbb{E}\left\|\nabla h_k(x_k^\ell) - \frac{1}{K}\sum_{k=1}^K\nabla h_k(x_k^\ell)\right\|^2}_{\text{Term IX}},$$

where $(a)$ utilizes Lemma C.1; $(b)$ results from the application of Lemma C.2; and $(c)$ results from Assumption 3.3.

Next, we bound Term IX below

$$\text{Term IX} := \mathbb{E}\left\|\nabla h_k(x_k^\ell) - \frac{1}{K}\sum_{k=1}^K\nabla h_k(x_k^\ell)\right\|^2$$

$$\overset{(a)}{\le} 3\mathbb{E}\left\|\nabla h_k(x_k^\ell) - \nabla h_k(\bar{x}^\ell)\right\|^2 + 3\mathbb{E}\left\|\frac{1}{K}\sum_{k=1}^K\left[\nabla h_k(\bar{x}^\ell) - \nabla h_k(x_k^\ell)\right]\right\|^2$$

$$+ 3\mathbb{E}\left\|\nabla h_k(\bar{x}^\ell) - \frac{1}{K}\sum_{k=1}^K\nabla h_k(\bar{x}^\ell)\right\|^2$$

$$\overset{(b)}{\le} 3L_h^2\mathbb{E}\left\|x_k^\ell - \bar{x}^\ell\right\|^2 + \frac{3L_h^2}{K}\sum_{k=1}^K\mathbb{E}\left\|x_k^\ell - \bar{x}^\ell\right\|^2 + 3\mathbb{E}\left\|\nabla h_k(\bar{x}^\ell) - \nabla h(\bar{x}^\ell)\right\|^2$$

$$\overset{(c)}{\le} 3L_h^2\mathbb{E}\left\|x_k^\ell - \bar{x}^\ell\right\|^2 + \frac{3L_h^2}{K}\sum_{k=1}^K\mathbb{E}\left\|x_k^\ell - \bar{x}^\ell\right\|^2 + 3\Delta_h^2,$$

where $(a)$ results from the application of Lemma C.1; $(b)$ utilizes Lipschitz smoothness of $h(\cdot)$ and the definition of $h(x) = \frac{1}{K}\sum_{k=1}^K h_k(x)$; finally, $(c)$ results from the bounded heterogeneity assumption Assumption 3.4. Substituting the bound on Term IX in the bound of Term VII, we get

$$\text{Term VII} \le \frac{2\sigma_h^2}{|b_{h_k}^t|} + 6L_h^2\mathbb{E}\left\|x_k^\ell - \bar{x}^\ell\right\|^2 + \frac{6L_h^2}{K}\sum_{k=1}^K\mathbb{E}\left\|x_k^\ell - \bar{x}^\ell\right\|^2 + 6\Delta_h^2.$$

Similarly, we bound Term VIII as

$$\text{Term VIII} := \mathbb{E}\left\|\frac{1}{|b_{g_k}^\ell|}\sum_{j\in b_{g_k}^\ell}\nabla g_k(x_k^\ell;\zeta_{k,j}^\ell) - \frac{1}{K}\sum_{k=1}^K\frac{1}{|b_{g_k}^\ell|}\sum_{j\in b_{g_k}^\ell}\nabla g_k(x_k^\ell;\zeta_{k,j}^\ell)\right\|^2$$

$$\stackrel{(a)}{\leq} 2\mathbb{E}\left\|\left[\frac{1}{|b_{g_k}^\ell|}\sum_{i\in b_{g_k}^\ell}\nabla g_k(x_k^\ell;\zeta_{k,i}^\ell) - \nabla g_k(x_k^\ell)\right] - \frac{1}{K}\sum_{k=1}^K\left[\frac{1}{|b_{g_k}^\ell|}\sum_{i\in b_{g_k}^\ell}\nabla g_k(x_k^\ell;\zeta_{k,i}^\ell) - \nabla g_k(x_k^\ell)\right]\right\|^2$$

$$+ 2\mathbb{E}\left\|\nabla g_k(x_k^\ell) - \frac{1}{K}\sum_{k=1}^K\nabla g_k(x_k^\ell)\right\|^2$$

$$\stackrel{(b)}{\leq} 2\mathbb{E}\left\|\frac{1}{|b_{g_k}^\ell|}\sum_{i\in b_{g_k}^\ell}\nabla g_k(x_k^\ell;\zeta_{k,i}^\ell) - \nabla g_k(x_k^\ell)\right\|^2 + 2\mathbb{E}\left\|\nabla g_k(x_k^\ell) - \frac{1}{K}\sum_{k=1}^K\nabla g_k(x_k^\ell)\right\|^2$$

$$\stackrel{(c)}{\leq} \frac{2\sigma_g^2}{|b_{g_k}^\ell|} + 2\underbrace{\mathbb{E}\left\|\nabla g_k(x_k^\ell) - \frac{1}{K}\sum_{k=1}^K\nabla g_k(x_k^\ell)\right\|^2}_{\text{Term X}},$$

where $(a)$ utilizes Lemma C.1; $(b)$ results from the application of Lemma C.2; and $(c)$ results from Assumption 3.3. Next, we bound Term X below

$$\text{Term X} := \mathbb{E}\left\|\nabla g_k(x_k^\ell) - \frac{1}{K}\sum_{k=1}^K\nabla g_k(x_k^\ell)\right\|^2$$

$$\stackrel{(a)}{\leq} 3\mathbb{E}\left\|\nabla g_k(x_k^\ell) - \nabla g_k(\bar{x}^\ell)\right\|^2 + 3\mathbb{E}\left\|\frac{1}{K}\sum_{k=1}^K\left[\nabla g_k(\bar{x}^\ell) - \nabla g_k(x_k^\ell)\right]\right\|^2$$

$$+ 3\mathbb{E}\left\|\nabla g_k(\bar{x}^\ell) - \frac{1}{K}\sum_{k=1}^K\nabla g_k(\bar{x}^\ell)\right\|^2$$

$$\stackrel{(b)}{\leq} 3L_g^2\mathbb{E}\left\|x_k^\ell - \bar{x}^\ell\right\|^2 + \frac{3L_g^2}{K}\sum_{k=1}^K\mathbb{E}\left\|x_k^\ell - \bar{x}^\ell\right\|^2 + 3\mathbb{E}\left\|\nabla g_k(\bar{x}^\ell) - \nabla g(\bar{x}^\ell)\right\|^2$$

$$\stackrel{(c)}{\leq} 3L_g^2\mathbb{E}\left\|x_k^\ell - \bar{x}^\ell\right\|^2 + \frac{3L_g^2}{K}\sum_{k=1}^K\mathbb{E}\left\|x_k^\ell - \bar{x}^\ell\right\|^2 + 3\Delta_g^2,$$

where $(a)$ results from the application of Lemma C.1; $(b)$ utilizes Lipschitz smoothness of $g(\cdot)$ and the definition of $g(x) = \frac{1}{K}\sum_{k=1}^K g_k(x)$; finally, $(c)$ results from the bounded heterogeneity assumption Assumption 3.4. Substituting the bound on Term X in the bound of Term VIII, we get

$$\text{Term VIII} \leq \frac{2\sigma_g^2}{|b_{g_k}^\ell|} + 6L_g^2\mathbb{E}\left\|x_k^\ell - \bar{x}^\ell\right\|^2 + \frac{6L_g^2}{K}\sum_{k=1}^K\mathbb{E}\left\|x_k^\ell - \bar{x}^\ell\right\|^2 + 6\Delta_g^2.$$

Next, we substitute the upper bounds on Terms VII and VIII in the expression of Term VI, we get

$$\text{Term VI} \leq \frac{4}{|b_{h_k}^\ell|}\sigma_h^2 + 12L_h^2\mathbb{E}\left\|x_k^\ell - \bar{x}^\ell\right\|^2 + \frac{12L_h^2}{K}\sum_{k=1}^K\mathbb{E}\left\|x_k^\ell - \bar{x}^\ell\right\|^2 + 12\Delta_h^2$$

$$+ \frac{4B_f^2}{|b_{g_k}^\ell|}\sigma_g^2 + 12B_f^2L_g^2\mathbb{E}\left\|x_k^\ell - \bar{x}^\ell\right\|^2 + \frac{12B_f^2L_g^2}{K}\sum_{k=1}^K\mathbb{E}\left\|x_k^\ell - \bar{x}^\ell\right\|^2 + 12B_f^2\Delta_g^2$$

$$= \left(12L_h^2 + 12B_f^2L_g^2\right)\mathbb{E}\left\|x_k^\ell - \bar{x}^\ell\right\|^2 + \left(\frac{12L_h^2 + 12B_f^2L_g^2}{K}\right)\sum_{k=1}^K\mathbb{E}\left\|x_k^\ell - \bar{x}^\ell\right\|^2$$

$$+ \frac{4}{|b_{h_k}^\ell|}\sigma_h^2 + \frac{4B_f^2}{|b_{g_k}^\ell|}\sigma_g^2 + 12\Delta_h^2 + 12B_f^2\Delta_g^2.$$

Therefore, we finally have the bound on Term IV as

$$\text{Term IV} \leq (I-1)\Big(24L_h^2 + 24B_f^2 L_g^2\Big) \sum_{\ell=t_s}^{t-1} \frac{(\eta^\ell)^2}{K} \sum_{k=1}^{K} \mathbb{E}\big\|x_k^\ell - \bar{x}^\ell\big\|^2$$

$$+ (I-1)\Big(\frac{4}{|b_h^t|}\sigma_h^2 + \frac{4B_f^2}{|b_g^t|}\sigma_g^2\Big)\sum_{\ell=t_s}^{t-1}(\eta^\ell)^2 + (I-1)\Big(12\Delta_h^2 + 12B_f^2\Delta_g^2\Big)\sum_{\ell=t_s}^{t-1}(\eta^\ell)^2.$$

where we have chosen $|b_{h_k}^\ell| = |b_h^t|$ and $|b_{g_k}^\ell| = |b_g^t|$ for all $k \in [K]$ and $\ell \in \{0, \ldots, T-1\}$.

Therefore, we have proof of the Lemma. $\qquad\square$

Next, we bound Term V from (18), we have

**Lemma F.5 (Descent in the estimate of $g(x)$).** *Under Assumptions 3.2-3.4, the iterates generated by Algorithm 2 satisfy:*

$$\mathbb{E}\Big\|\bar{y}^t - \frac{1}{K}\sum_{k=1}^{K}g_k(x_k^t)\Big\|^2$$

$$\leq (1-\beta^t)^2\mathbb{E}\Big\|\bar{y}^{t-1} - \frac{1}{K}\sum_{k=1}^{K}g_k(x_k^{t-1})\Big\|^2 + \frac{8(\eta^t)^2(1-\beta^t)^2 B_g^2}{|b_g|K}\mathbb{E}\Big\|\frac{1}{K}\sum_{k=1}^{K}\mathbb{E}[\Phi_k(x_k^t; \bar{\xi}_k^t)|\mathcal{F}^t]\Big\|^2$$

$$+ \frac{(\eta^t)^2(1-\beta^t)^2 B_g^2(96L_h^2 + 96B_f^2 L_g^2)}{|b_g|K^2}\sum_{k=1}^{K}\mathbb{E}\big\|x_k^t - \bar{x}^t\big\|^2 + \frac{4(\eta^t)^2(1-\beta^t)^2 B_g^2}{|b_h|K}\sigma_h^2$$

$$+ \frac{2(\beta^t)^2 + 4(\eta^t)^2(1-\beta^t)^2 B_g^2 B_f^2}{|b_g|K}\sigma_g^2 + \frac{48(\eta^t)^2(1-\beta^t)^2 B_g^2}{|b_g|K}\Delta_h^2 + \frac{48(\eta^t)^2(1-\beta^t)^2 B_f^2 B_g^2}{|b_g|K}\Delta_g^2.$$

*where we have chosen $|b_h^t| = |b_h|$ and $|b_{g_k}^t| = |b_g|$ for all $k \in [K]$ and $t \in [T]$.*

*Proof.* From the definition of Term V, we have

$$\text{Term V} := \mathbb{E}\Big\|\bar{y}^{t+1} - \frac{1}{K}\sum_{k=1}^{K}g_k(x_k^{t+1})\Big\|^2$$

$$\overset{(a)}{=} \mathbb{E}\Big\|\frac{1}{K}\sum_{k=1}^{K}\big[y_k^{t+1} - g_k(x_k^{t+1})\big]\Big\|^2$$

$$\overset{(b)}{=} \mathbb{E}\Big\|\frac{1}{K}\sum_{k=1}^{K}\Big[(1-\beta^{t+1})\Big(y_k^t + \frac{1}{|b_{g_k}^{t+1}|}\sum_{i\in b_{g_k}^{t+1}}g_k(x_k^{t+1};\zeta_{k,i}^{t+1}) - \frac{1}{|b_{g_k}^{t+1}|}\sum_{i\in b_{g_k}^{t+1}}g_k(x_k^t;\zeta_{k,i}^{t+1})\Big)$$

$$+ \frac{\beta^{t+1}}{|b_{g_k}^{t+1}|}\sum_{i\in b_{g_k}^{t+1}}g_k(x_k^{t+1},\zeta_{k,i}^{t+1}) - g_k(x_k^{t+1})\Big]\Big\|^2$$

$$\overset{(c)}{=} (1-\beta^{t+1})^2\,\mathbb{E}\Big\|\frac{1}{K}\sum_{k=1}^{K}\big[y_k^t - g_k(x_k^t)\big]\Big\|^2$$

$$+ \mathbb{E}\Big\|\frac{1}{K}\sum_{k=1}^{K}\Big[(1-\beta^{t+1})\Big[(g_k(x_k^t) - g_k(x_k^{t+1})) - \frac{1}{|b_{g_k}^{t+1}|}\sum_{i\in b_{g_k}^{t+1}}\big(g_k(x_k^t;\zeta_{k,i}^{t+1}) - g_k(x_k^{t+1};\zeta_{k,i}^{t+1})\big)\Big]$$

$$+ \beta^{t+1}\Big(\frac{1}{|b_{g_k}^{t+1}|}\sum_{i\in b_{g_k}^{t+1}}g_k(x_k^{t+1};\zeta_{k,i}^{t+1}) - g_k(x_k^{t+1})\Big)\Big]\Big\|^2$$

$$\overset{(d)}{\leq} (1-\beta^{t+1})^2 \, \mathbb{E}\left\|\bar{y}^t - \frac{1}{K}\sum_{k=1}^{K} g_k(x_k^t)\right\|^2 + \frac{2(\beta^{t+1})^2}{|b_g|K}\sigma_g^2$$

$$+ \frac{2(1-\beta^{t+1})^2}{K^2}\sum_{k=1}^{K}\frac{1}{|b_g|^2}\sum_{i\in b_{g_k}^{t+1}} \mathbb{E}\left\|\left(g_k(x_k^t)-g(x_k^{t+1})\right)-\left(g_k(x_k^t;\zeta_{k,i}^{t+1})-g_k(x_k^{t+1};\zeta_{k,i}^{t+1})\right)\right\|^2$$

$$\overset{(e)}{\leq} (1-\beta^{t+1})^2 \, \mathbb{E}\left\|\bar{y}^t - \frac{1}{K}\sum_{k=1}^{K} g_k(x_k^t)\right\|^2 + \frac{2(\beta^{t+1})^2}{|b_g|K}\sigma_g^2$$

$$+ \frac{2(1-\beta^{t+1})^2}{K^2}\sum_{k=1}^{K}\frac{1}{|b_g|^2}\sum_{i\in b_{g_k}^{t+1}} \mathbb{E}\left\|g_k(x_k^t;\zeta_{k,i}^{t+1})-g_k(x_k^{t+1};\zeta_{k,i}^{t+1})\right\|^2$$

$$\overset{(f)}{\leq} (1-\beta^{t+1})^2 \, \mathbb{E}\left\|\bar{y}^t - \frac{1}{K}\sum_{k=1}^{K} g_k(x_k^t)\right\|^2 + \frac{2(\beta^{t+1})^2}{|b_g|K}\sigma_g^2 + \frac{2(1-\beta^{t+1})^2 B_g^2}{|b_g|K^2}\sum_{k=1}^{K}\mathbb{E}\left\|x_k^{t+1}-x_k^t\right\|^2$$

$$\overset{(g)}{\leq} (1-\beta^{t+1})^2\mathbb{E}\left\|\bar{y}^t - \frac{1}{K}\sum_{k=1}^{K} g_k(x_k^t)\right\|^2 + \frac{2(\beta^{t+1})^2}{|b_g|K}\sigma_g^2$$

$$+ \frac{2(\eta^t)^2(1-\beta^{t+1})^2 B_g^2}{|b_g|K^2}\sum_{k=1}^{K}\underbrace{\mathbb{E}\left\|\nabla\Phi_k(x_k^t;\bar{\xi}_k^t)\right\|^2}_{\text{Term XI}},$$

where $(a)$ follows from the definition of $\bar{y}^{t+1}$; $(b)$ uses the update rule (8) for $y_k^{t+1}$; $(c)$ results from adding and subtracting $(1-\beta^{t+1})g_k(x_k^t)$ and utilizing the fact that the second term in the expression has zero-mean which follows from Assumption 3.3; $(d)$ uses Young's inequality, Assumption 3.3 and by choosing $|b_h^t| = |b_h|$ and $|b_{g_k}^t| = |b_g|$ for all $k\in[K]$ and $t\in[T]$; $(e)$ results from the fact that for a random variable $X$, we have $\mathbb{E}\|X-\mathbb{E}[X]\|^2 \leq \mathbb{E}\|X\|^2$; $(f)$ uses the mean-squared Lipschitzness of $g_k(\cdot)$ in Assumption 3.2; finally $(g)$ results from the update rule of Algorithm 2.

Next, we bound Term XI below

$$\text{Term XI} \coloneqq \mathbb{E}\left\|\nabla\Phi_k(x_k^t;\bar{\xi}_k^t)\right\|^2$$

$$\overset{(a)}{\leq} 2\mathbb{E}\left\|\nabla\Phi_k(x_k^t;\bar{\xi}_k^t)-\mathbb{E}[\nabla\Phi_k(x_k^t;\bar{\xi}_k^t)|\mathcal{F}^t]\right\|^2 + 2\mathbb{E}\left\|\mathbb{E}[\nabla\Phi_k(x_k^t;\bar{\xi}_k^t)|\mathcal{F}^t]\right\|^2$$

$$\overset{(b)}{\leq} \frac{2\sigma_h^2}{|b_h|} + \frac{2\sigma_g^2 B_f^2}{|b_g|} + 4\,\mathbb{E}\underbrace{\left\|\mathbb{E}[\nabla\Phi_k(x_k^t;\bar{\xi}_k^t)|\mathcal{F}^t]-\frac{1}{K}\sum_{k=1}^{K}\mathbb{E}[\nabla\Phi_k(x_k^t;\bar{\xi}_k^t)|\mathcal{F}^t]\right\|^2}_{\text{Term XII}}$$

$$+ 4\mathbb{E}\left\|\frac{1}{K}\sum_{k=1}^{K}\mathbb{E}[\nabla\Phi_k(x_k^t;\bar{\xi}_k^t)|\mathcal{F}^t]\right\|^2,$$

where $(a)$ results from the application of Young's inequality and $(b)$ results from Assumptions 3.2 and 3.3 along with the application of Young's inequality.

Next, we bound Term XII in the above expression.

$$\text{Term XII} \coloneqq \mathbb{E}\left\|\mathbb{E}[\nabla\Phi_k(x_k^t;\bar{\xi}_k^t)|\mathcal{F}^t]-\frac{1}{K}\sum_{k=1}^{K}\mathbb{E}[\nabla\Phi_k(x_k^t;\bar{\xi}_k^t)|\mathcal{F}^t]\right\|^2$$

$$\overset{(a)}{=} \mathbb{E}\left\|\nabla h_k(x_k^t)+\nabla g_k(x_k^t)\nabla f(\bar{y}^t)-\left[\frac{1}{K}\sum_{k=1}^{K}\left(\nabla h_k(x_k^t)+\nabla g_k(x_k^t)\nabla f(\bar{y}^t)\right)\right]\right\|^2$$

$$\overset{(b)}{\leq} 2\mathbb{E}\left\|\nabla h_k(x_k^t)-\frac{1}{K}\sum_{k=1}^{K}\nabla h_k(x_k^t)\right\|^2 + 2\mathbb{E}\left\|\nabla g_k(x_k^t)\nabla f(\bar{y}^t)-\frac{1}{K}\sum_{k=1}^{K}\nabla g_k(x_k^t)\nabla f(\bar{y}^t)\right]\right\|^2$$

$$\overset{(c)}{\leq} 2\,\mathbb{E}\left\|\nabla h_k(x_k^t) - \frac{1}{K}\sum_{k=1}^{K}\nabla h_k(x_k^t)\right\|^2 + \underbrace{2B_f^2\,\mathbb{E}\left\|\left\|\nabla g_k(x_k^t) - \frac{1}{K}\sum_{k=1}^{K}\nabla g_k(x_k^t)\right]\right\|^2}_{\text{Term X}}$$

$$\overset{(d)}{\leq} (6L_h^2 + 6B_f^2 L_g^2)\mathbb{E}\|x_k^t - \bar{x}^t\|^2 + \frac{6L_h^2 + 6B_f^2 L_g^2}{K}\sum_{k=1}^{K}\mathbb{E}\|x_k^t - \bar{x}^t\|^2 + 6\Delta_h^2 + 6B_f^2\Delta_g^2$$

where $(a)$ above uses the definition of $\nabla\Phi_k(x_k^t;\bar{\xi}_k^t)$ in (7) and Assumption 3.3; $(b)$ results from the application of Young's inequality; $(c)$ utilized Assumtion 3.2; finally, $(d)$ results from the application of Assumptions 3.2 and 3.4.

Replacing in the upper bound for Term XI, we get

$$\text{Term XI} \leq 4\mathbb{E}\left\|\frac{1}{K}\sum_{k=1}^{K}\mathbb{E}[\Phi_k(x_k^t;\bar{\xi}_k^t)|\mathcal{F}^t]\right\|^2 + (24L_h^2 + 24B_f^2 L_g^2)\mathbb{E}\|x_k^t - \bar{x}^t\|^2$$

$$+ \frac{24L_h^2 + 24B_f^2 L_g^2}{K}\sum_{k=1}^{K}\mathbb{E}\|x_k^t - \bar{x}^t\|^2 + \frac{2\sigma_h^2}{|b_h|} + \frac{2\sigma_g^2 B_f^2}{|b_g|} + 24\Delta_h^2 + 24B_f^2\Delta_g^2.$$

Substituting the bound on Term XI in the bound of Term V, we get

$$\mathbb{E}\left\|\bar{y}^{t+1} - \frac{1}{K}\sum_{k=1}^{K}g_k(x_k^{t+1})\right\|^2$$

$$\leq (1-\beta^{t+1})^2\,\mathbb{E}\left\|\bar{y}^t - \frac{1}{K}\sum_{k=1}^{K}g_k(x_k^t)\right\|^2 + \frac{8(\eta^t)^2(1-\beta^{t+1})^2 B_g^2}{|b_g|K}\mathbb{E}\left\|\frac{1}{K}\sum_{k=1}^{K}\mathbb{E}[\Phi_k(x_k^t;\bar{\xi}_k^t)|\mathcal{F}^t]\right\|^2$$

$$+ \frac{(\eta^t)^2(1-\beta^{t+1})^2 B_g^2(96L_h^2 + 96B_f^2 L_g^2)}{|b_g|K^2}\sum_{k=1}^{K}\mathbb{E}\|x_k^t - \bar{x}^t\|^2 + \frac{4(\eta^t)^2(1-\beta^{t+1})^2 B_g^2}{|b_h|K}\sigma_h^2$$

$$+ \frac{2(\beta^{t+1})^2 + 4(\eta^t)^2(1-\beta^{t+1})^2 B_g^2 B_f^2}{|b_g|K}\sigma_g^2 + \frac{48(\eta^t)^2(1-\beta^{t+1})^2 B_g^2}{|b_g|K}\Delta_h^2 + \frac{48(\eta^t)^2(1-\beta^{t+1})^2 B_f^2 B_g^2}{|b_g|K}\Delta_g^2.$$

Therefore, we have proof of Lemma. $\qquad\square$

Next, we show descent in the potential function specially designed to show convergence of Algorithm 2. For this purpose, we define the potential function as

$$V^t = \mathbb{E}[\Phi(\bar{x}^t)] + \mathbb{E}\left\|\bar{y}^t - \frac{1}{K}\sum_{k=1}^{K}g_k(x_k^t)\right\|^2. \tag{21}$$

Next, we derive the descent in the potential function.

**Lemma F.6 (Descent in Potential Function).** *Under Assumptions 3.2-3.4 with the choice of momentum-parameter $\beta^{t+1} = c_\beta\eta^t$ with $c_\beta = 4B_g^4 L_f^2$ where step-size $\eta^t$ is chosen such that*

$$\eta^t \leq \left\{\frac{|b_g|K}{2(L_\Phi|b_g|K + 8B_g^2)}, \frac{|b_g|K\left(L_h^2 + 2B_f^2 L_g^2 + 4B_g^4 L_f^2\right)}{B_g^2\left(96L_h^2 + 96B_f^2 L_g^2\right)}\right\}$$

*the iterates generated by Algorithm 2 satisfy*

$$V^{t+1} - V^t \leq -\frac{\eta^t}{2}\mathbb{E}\|\nabla\Phi(\bar{x}^t)\|^2 + \eta^t\left(2L_h^2 + 4B_f^2 L_g^2 + 8B_g^4 L_F^2\right)\frac{1}{K}\sum_{k=1}^{K}\mathbb{E}\|x_k^t - \bar{x}^t\|^2$$

$$+ \frac{2(\eta^t)^2 L_\Phi}{K|b_h|}\sigma_h^2 + \frac{4(\eta^t)^2 B_g^2}{|b_h|K}\sigma_h^2 + \frac{2(\eta^t)^2 L_\Phi B_f^2}{|b_g|K}\sigma_g^2 + \frac{(\eta^t)^2(2c_\beta^2 + 4B_g^2 B_f^2)}{|b_g|K}\sigma_g^2$$

$$+ \frac{48(\eta^t)^2 B_g^2}{|b_g|K}\Delta_h^2 + \frac{48(\eta^t)^2 B_f^2 B_g^2}{|b_g|K}\Delta_g^2.$$

*Proof.* From the definition of $V^t$ in (21) and using Lemmas F.3 and F.5, we get

$$V^{t+1} - V^t = \mathbb{E}[\Phi(\bar{x}^{t+1}) - \Phi(\bar{x}^t)] + \mathbb{E}\left\|\bar{y}^{t+1} - \frac{1}{K}\sum_{k=1}^{K} g_k(x_k^{t+1})\right\|^2 - \mathbb{E}\left\|\bar{y}^t - \frac{1}{K}\sum_{k=1}^{K} g_k(x_k^t)\right\|^2$$

$$\leq -\frac{\eta^t}{2}\mathbb{E}\big\|\nabla\Phi(\bar{x}^t)\big\|^2 - \left(\frac{\eta^t}{2} - (\eta^t)^2 L_\Phi - \frac{8(\eta^t)^2 B_g^2}{|b_g|K}\right)\mathbb{E}\left\|\frac{1}{K}\sum_{k=1}^{K}\mathbb{E}\big[\nabla\Phi_k(x_k^t;\bar{\xi}_k^t)\big|\mathcal{F}^t\big]\right\|^2$$

$$+ \left(\eta^t\big(L_h^2 + 2B_f^2 L_g^2 + 4B_g^4 L_F^2\big) + \frac{(\eta^t)^2 B_g^2(96L_h^2 + 96B_f^2 L_g^2)}{|b_g|K}\right)\frac{1}{K}\sum_{k=1}^{K}\mathbb{E}\|x_k^t - \bar{x}^t\|^2$$

$$+ \big(4B_g^4 L_f^2 \eta^t - \beta^{t+1}\big)\mathbb{E}\left\|\bar{y}^t - \frac{1}{K}\sum_{k=1}^{K} g_k(x_k^t)\right\|^2 + \frac{2(\eta^t)^2 L_\Phi}{K|b_h|}\sigma_h^2 + \frac{4(\eta^t)^2 B_g^2}{|b_h|K}\sigma_h^2$$

$$+ \frac{2(\eta^t)^2 L_\Phi B_f^2}{|b_g|K}\sigma_g^2 + \frac{2(\beta^{t+1})^2 + 4(\eta^t)^2 B_g^2 B_f^2}{|b_g|K}\sigma_g^2 + \frac{48(\eta^t)^2 B_g^2}{|b_g|K}\Delta_h^2 + \frac{48(\eta^t)^2 B_f^2 B_g^2}{|b_g|K}\Delta_g^2$$

$$\overset{(a)}{\leq} -\frac{\eta^t}{2}\mathbb{E}\big\|\nabla\Phi(\bar{x}^t)\big\|^2 + \eta^t\big(2L_h^2 + 4B_f^2 L_g^2 + 8B_g^4 L_F^2\big)\frac{1}{K}\sum_{k=1}^{K}\mathbb{E}\|x_k^t - \bar{x}^t\|^2$$

$$+ \frac{2(\eta^t)^2 L_\Phi}{K|b_h|}\sigma_h^2 + \frac{4(\eta^t)^2 B_g^2}{|b_h|K}\sigma_h^2 + \frac{2(\eta^t)^2 L_\Phi B_f^2}{|b_g|K}\sigma_g^2 + \frac{(\eta^t)^2(2c_\beta^2 + 4B_g^2 B_f^2)}{|b_g|K}\sigma_g^2$$

$$+ \frac{48(\eta^t)^2 B_g^2}{|b_g|K}\Delta_h^2 + \frac{48(\eta^t)^2 B_f^2 B_g^2}{|b_g|K}\Delta_g^2.$$

where $(a)$ results from the choice of $\beta^t$ and $\eta_t$ given in the statement of the Lemma.

Therefore, we have the proof. $\qquad\square$

**Theorem F.7** (**Potential Function**). *Under Assumptions 3.2-3.4 and the choice of step-size $\eta^t = \eta$ such that we have*

$$\eta \leq \frac{1}{3I\big(24L_h^2 + 24B_f^2 L_g^2\big)^{1/2}}$$

*the iterates generated by Algorithm 2 satisfy*

$$V^T - V^0 \leq -\frac{\eta}{2}\sum_{t=0}^{T-1}\mathbb{E}\big\|\nabla\Phi(\bar{x}^t)\big\|^2 + \eta^3(I-1)^2\frac{\big(10L_h^2 + 20B_f^2 L_g^2 + 40B_g^4 L_F^2\big)}{|b_h|}\sigma_h^2\, T$$

$$+ \frac{2\eta^2 L_\Phi}{K|b_h|}\sigma_h^2\, T + \frac{4\eta^2 B_g^2}{|b_h|K}\sigma_h^2\, T + \eta^3(I-1)^2\frac{\big(10B_f^2 L_h^2 + 20B_f^4 L_g^2 + 40B_f^2 B_g^4 L_F^2\big)}{|b_g|}\sigma_g^2\, T$$

$$+ \frac{2\eta^2 L_\Phi B_f^2}{|b_g|K}\sigma_g^2\, T + \frac{\eta^2(2c_\beta^2 + 4B_g^2 B_f^2)}{|b_g|K}\sigma_g^2\, T + \eta^3(I-1)^2\big(30L_h^2 + 60B_f^2 L_g^2 + 120B_g^4 L_F^2\big)\Delta_h^2\, T$$

$$+ \frac{48\eta^2 B_g^2}{|b_g|K}\Delta_h^2\, T + \eta^3(I-1)^2\big(30B_f^2 L_h^2 + 60B_f^4 L_g^2 + 120B_f^2 B_g^4 L_F^2\big)\Delta_g^2\, T + \frac{48\eta^2 B_f^2 B_g^2}{|b_g|K}\Delta_g^2\, T.$$

*Proof.* Telescoping the sum of Lemma F.6 for $t = \{0, 1, \ldots, T-1\}$, we get

$$V^T - V^0 \leq -\frac{\eta}{2}\sum_{t=0}^{T-1}\mathbb{E}\big\|\nabla\Phi(\bar{x}^t)\big\|^2 + \eta\big(2L_h^2 + 4B_f^2 L_g^2 + 8B_g^4 L_F^2\big)\underbrace{\sum_{t=0}^{T-1}\frac{1}{K}\sum_{k=1}^{K}\mathbb{E}\|x_k^t - \bar{x}^t\|^2}_{\text{Term XIII}}$$

$$+ \frac{2\eta^2 L_\Phi}{K|b_h|}\sigma_h^2\, T + \frac{4\eta^2 B_g^2}{|b_h|K}\sigma_h^2\, T + \frac{2\eta^2 L_\Phi B_f^2}{|b_g|K}\sigma_g^2\, T + \frac{\eta^2(2c_\beta^2 + 4B_g^2 B_f^2)}{|b_g|K}\sigma_g^2\, T$$

$$+ \frac{48\eta^2 B_g^2}{|b_g|K}\Delta_h^2\, T + \frac{48\eta^2 B_f^2 B_g^2}{|b_g|K}\Delta_g^2\, T. \qquad (22)$$

We bound Term XIII in (22) using Lemma (F.4). Note that we have from Lemma (F.4)

$$\frac{1}{K}\sum_{k=1}^{K}\mathbb{E}\|x_k^t - \bar{x}^t\|^2 \leq (I-1)\Big(24L_h^2 + 24B_f^2 L_g^2\Big)\sum_{\ell=t_s}^{t-1}\frac{(\eta^\ell)^2}{K}\sum_{k=1}^{K}\mathbb{E}\|x_k^\ell - \bar{x}^\ell\|^2$$

$$+ (I-1)\Big(\frac{4}{|b_h^t|}\sigma_h^2 + \frac{4B_f^2}{|b_g^t|}\sigma_g^2\Big)\sum_{\ell=t_s}^{t-1}(\eta^\ell)^2 + (I-1)\Big(12\Delta_h^2 + 12B_f^2\Delta_g^2\Big)\sum_{\ell=t_s}^{t-1}(\eta^\ell)^2$$

Summing the above from $t = t_s$ to $t_{s+1} - 1$, we get

$$\sum_{t=t_s}^{t_{s+1}-1}\frac{1}{K}\sum_{k=1}^{K}\mathbb{E}\|x_k^t - \bar{x}^t\|^2 \overset{(a)}{\leq} \eta^2(I-1)\Big(24L_h^2 + 24B_f^2 L_g^2\Big)\sum_{t=t_s}^{t_{s+1}-1}\sum_{\ell=t_s}^{t-1}\frac{1}{K}\sum_{k=1}^{K}\mathbb{E}\|x_k^\ell - \bar{x}^\ell\|^2$$

$$+ \eta^2(I-1)^2 I\Big(\frac{4}{|b_h^t|}\sigma_h^2 + \frac{4B_f^2}{|b_g^t|}\sigma_g^2\Big) + \eta^2(I-1)^2 I\Big(12\Delta_h^2 + 12B_f^2\Delta_g^2\Big)$$

$$\overset{(b)}{\leq} \eta^2(I-1)\Big(24L_h^2 + 24B_f^2 L_g^2\Big)\sum_{t=t_s}^{t_{s+1}-1}\sum_{\ell=t_s}^{t_{s+1}-1}\frac{1}{K}\sum_{k=1}^{K}\mathbb{E}\|x_k^\ell - \bar{x}^\ell\|^2$$

$$+ \eta^2(I-1)^2 I\Big(\frac{4}{|b_h^t|}\sigma_h^2 + \frac{4B_f^2}{|b_g^t|}\sigma_g^2\Big) + \eta^2(I-1)^2 I\Big(12\Delta_h^2 + 12B_f^2\Delta_g^2\Big)$$

$$\overset{(c)}{\leq} \eta^2(I-1)I\Big(24L_h^2 + 24B_f^2 L_g^2\Big)\sum_{t=t_s}^{t_{s+1}-1}\frac{1}{K}\sum_{k=1}^{K}\mathbb{E}\|x_k^t - \bar{x}^t\|^2$$

$$+ \eta^2(I-1)^2 I\Big(\frac{4}{|b_h^t|}\sigma_h^2 + \frac{4B_f^2}{|b_g^t|}\sigma_g^2\Big) + \eta^2(I-1)^2 I\Big(12\Delta_h^2 + 12B_f^2\Delta_g^2\Big)$$

where in $(a)$ we have used the fact that $\eta^t = \eta$ for all $t \in [T]$ and $(t-1) - t_s \leq I - 1$ for $t \in [t_s, t_{s+1} - 1]$; $(b)$ results from the fact that $t \leq t_{s+1}$; finally, $(c)$ again uses the fact that $(t-1) - t_s \leq I - 1$ for $t \in [t_s, t_{s+1} - 1]$.

Summing the above from $s = \{0, 1, \ldots, S\}$ and using the fact that $S \times I = T - 1$, we get

$$\sum_{t=0}^{T-1}\frac{1}{K}\sum_{k=1}^{K}\mathbb{E}\|x_k^t - \bar{x}^t\|^2 \leq \eta^2 I^2\Big(24L_h^2 + 24B_f^2 L_g^2\Big)\sum_{t=0}^{T-1}\frac{1}{K}\sum_{k=1}^{K}\mathbb{E}\|x_k^t - \bar{x}^t\|^2$$

$$+ \eta^2(I-1)^2\Big(\frac{4}{|b_h^t|}\sigma_h^2 + \frac{4B_f^2}{|b_g^t|}\sigma_g^2\Big)T + \eta^2(I-1)^2\Big(12\Delta_h^2 + 12B_f^2\Delta_g^2\Big)T.$$

Rearranging the terms, we get

$$\Big(1 - \eta^2 I^2\big(24L_h^2 + 24B_f^2 L_g^2\big)\Big)\sum_{t=0}^{T-1}\frac{1}{K}\sum_{k=1}^{K}\mathbb{E}\|x_k^t - \bar{x}^t\|^2 \leq \eta^2(I-1)^2\Big(\frac{4}{|b_h^t|}\sigma_h^2 + \frac{4B_f^2}{|b_g^t|}\sigma_g^2\Big)T$$

$$+ \eta^2(I-1)^2\Big(12\Delta_h^2 + 12B_f^2\Delta_g^2\Big)T.$$

Finally, choosing $\eta \leq \frac{1}{3I\big(24L_h^2 + 24B_f^2 L_g^2\big)^{1/2}}$, such that we have $1 - \eta^2 I^2\big(24L_h^2 + 24B_f^2 L_g^2\big) \geq 8/9$, utilizing this we get

$$\text{Term XIII} := \sum_{t=0}^{T-1}\frac{1}{K}\sum_{k=1}^{K}\mathbb{E}\|x_k^t - \bar{x}^t\|^2$$

$$\leq \eta^2(I-1)^2\Big(\frac{5}{|b_h^t|}\sigma_h^2 + \frac{5B_f^2}{|b_g^t|}\sigma_g^2\Big)T + \eta^2(I-1)^2\Big(15\Delta_h^2 + 15B_f^2\Delta_g^2\Big)T.$$

Finally, substituting the bound on Term XIII in (22), we get

$$
V^T - V^0 \leq -\frac{\eta}{2}\sum_{t=0}^{T-1}\mathbb{E}\big\|\nabla\Phi(\bar{x}^t)\big\|^2 + \eta^3(I-1)^2\frac{\big(10L_h^2 + 20B_f^2L_g^2 + 40B_g^4L_F^2\big)}{|b_h|}\sigma_h^2\,T
$$

$$
+ \frac{2\eta^2 L_\Phi}{K|b_h|}\sigma_h^2\,T + \frac{4\eta^2 B_g^2}{|b_h|K}\sigma_h^2\,T + \eta^3(I-1)^2\frac{\big(10B_f^2L_h^2 + 20B_f^4L_g^2 + 40B_f^2B_g^4L_F^2\big)}{|b_g|}\sigma_g^2\,T
$$

$$
+ \frac{2\eta^2 L_\Phi B_f^2}{|b_g|K}\sigma_g^2\,T + \frac{\eta^2(2c_\beta^2 + 4B_f^2B_g^2)}{|b_g|K}\sigma_g^2\,T + \eta^3(I-1)^2\big(30L_h^2 + 60B_f^2L_g^2 + 120B_g^4L_F^2\big)\Delta_h^2\,T
$$

$$
+ \frac{48\eta^2 B_g^2}{|b_g|K}\Delta_h^2\,T + \eta^3(I-1)^2\big(30B_f^2L_h^2 + 60B_f^4L_g^2 + 120B_f^2B_g^4L_F^2\big)\Delta_g^2\,T + \frac{48\eta^2 B_f^2B_g^2}{|b_g|K}\Delta_g^2\,T.
$$

Therefore, we have the proof. $\qquad\square$

Now, we are finally ready to prove Theorem 5.1.

*Proof.* Assuming $|b_h| = |b_g| = |b|$ and defining $\bar{L}_{f,g} := 10L_h^2 + B_f^2L_g^2 + 40B_g^4L_f^2$. Rearranging the terms in the expression of Theorem F.7 and multiplying both sides by $2/\eta T$ we get

$$
\frac{1}{T}\sum_{t=0}^{T-1}\mathbb{E}\big\|\nabla\Phi(\bar{x}^t)\big\|^2 \leq \frac{2\big[\Phi(\bar{x}^0) - \Phi(x^*) + \big\|\bar{y}^0 - g(\bar{x}^0)\big\|^2\big]}{\eta T} + \eta^2(I-1)^2\bigg[\frac{2\bar{L}_{f,g}}{|b|}\sigma_h^2 + \frac{2B_f^2\bar{L}_{f,g}}{|b|}\sigma_g^2\bigg]
$$

$$
+ \eta^2(I-1)^2\bigg[6\bar{L}_{f,g}\Delta_h^2 + 6B_f^2\bar{L}_{f,g}\Delta_g^2\bigg] + \eta\bigg[\frac{4L_\Phi + 8B_g^2}{|b|K}\sigma_h^2 + \frac{4L_\Phi B_f^2 + 4c_\beta^2 + 8B_f^2B_g^2}{|b|K}\sigma_g^2\bigg]
$$

$$
+ \eta\bigg[\frac{96B_g^2}{|b|K}\Delta_h^2 + \frac{96B_f^2B_g^2}{|b|K}\Delta_g^2\bigg],
$$

where the first term on the right follows from the fact that $\Phi(\bar{x}^T) \geq \Phi(x^*)$ and $\big\|\bar{y}^T - 1/K\sum_{k=1}^K g_k(x_k^T)\big\|^2 \geq 0$.

Next, choosing $\eta = \sqrt{\frac{|b|K}{T}}$ then for $T \geq \big(216L_h^2 + 216B_f^2L_g^2\big)I^2|b|K$ such that $\eta \leq \frac{1}{3I\big(24L_h^2 + 24B_f^2L_g^2\big)^{1/2}}$ in Theorem F.7 is satisfied, we get the following

$$
\frac{1}{T}\sum_{t=0}^{T-1}\mathbb{E}\big\|\nabla\Phi(\bar{x}^t)\big\|^2 \leq \frac{2\big[\Phi(\bar{x}^0) - \Phi(x^*) + \big\|\bar{y}^0 - g(\bar{x}^0)\big\|^2\big]}{\sqrt{|b|KT}} + \frac{K(I-1)^2}{T}\bigg[2\bar{L}_{f,g}\sigma_h^2 + 2B_f^2\bar{L}_{f,g}\sigma_g^2\bigg]
$$

$$
+ \frac{|b|K(I-1)^2}{T}\bigg[6\bar{L}_{f,g}\Delta_h^2 + 6B_f^2\bar{L}_{f,g}\Delta_g^2\bigg] + \frac{1}{\sqrt{|b|KT}}\bigg[\big(4L_\Phi + 8B_g^2\big)\sigma_h^2 + \big(4L_\Phi B_f^2 + 4c_\beta^2 + 8B_f^2B_g^2\big)\sigma_g^2\bigg]
$$

$$
+ \frac{1}{\sqrt{|b|KT}}\bigg[96B_g^2\,\Delta_h^2 + 96B_f^2B_g^2\,\Delta_g^2\bigg],
$$

Explicitly choosing $I = T^{1/4}/(|b|K)^{3/4}$, we get

$$
\mathbb{E}\big\|\nabla\Phi(\bar{x}^{a(T)})\big\|^2 \leq \frac{2\big[\Phi(\bar{x}^0) - \Phi(x^*) + \big\|\bar{y}^0 - g(\bar{x}^0)\big\|^2\big]}{\sqrt{|b|KT}} + \frac{C_{\sigma_h}}{\sqrt{|b|KT}}\sigma_h^2 + \frac{C_{\sigma_g}}{\sqrt{|b|KT}}\sigma_g^2
$$

$$
+ \frac{C_{\Delta_h}}{\sqrt{|b|KT}}\Delta_h^2 + \frac{C_{\Delta_g}}{\sqrt{|b|KT}}\Delta_g^2.
$$

where the constants $C_{\sigma_h}, C_{\sigma_g}, C_{\Delta_f}$, and $C_{\Delta_g}$ are defined as:

$$
C_{\sigma_h} = 2\bar{L}_{f,g} + 4L_\Phi + 8B_g^2
$$

$$
C_{\sigma_g} = 2B_f^2\bar{L}_{f,g} + 4L_\Phi B_f^2 + 4c_\beta^2 + 8B_f^2B_g^2
$$

$$
C_{\Delta_f} = 6\bar{L}_{f,g} + 96B_g^2
$$

$$
C_{\Delta_g} = 6B_f^2\bar{L}_{f,g} + 96B_f^2B_g^2.
$$

The constant $c_\beta$ is defined in the statement of Lemma F.6.

Hence, Theorem 5.1 is proved. □