# OpenReview forum: "FedDRO: Federated Compositional Optimization for Distributionally Robust Learning"
_ICLR.cc/2024/Conference — ICLR 2024 Conference Withdrawn Submission_

### Official Review · Reviewer_J3Ha · 2023-10-28

**Soundness:** 3 good
**Presentation:** 3 good
**Contribution:** 2 fair
**Rating:** 5
**Confidence:** 4

**Summary:**

This paper investigate an interesting problem of federated compositional optimization and propose a novel CO setting, which is more general compared to previous studies. Furthermore, the authors also present the FedDRO algorithm with provable theoretical guarantees. Empirical studies are also conducted.

**Strengths:**

1. This paper is well-originazed and clearly written.

2. The paper investigates an important problem and introduce a general federated compositional optimization setting.

3. Experimental results are provided to verify the performance of the proposed algorithms

**Weaknesses:**

1. The authors have introduced a general setting for DRO problems, including DRO with KL-Divergence and $\chi^2$-Divergence. Are there other DRO optimization problems that fit this setting?

2. The exact form of the optimization problem in two experiments should be specified.

3. How is the dataset distributed across each client? Does it reflect the diversity in data distribution?

4. Utilizing the small batch size is one strength of FedDRO. However, this isn't evident in the experiments. Specifically, |b|=16 in FedDRO, while |b|=4 (or 8) in GCIVR (with the client count m=8). Therefore, experiments should be conducted with small batch size or more clients (ensuring large batch size in GCIVR).

5. The experiments are repeated only five times, and it would further strengthen the reliability if reruning the experiments at least ten times. Furthermore, presenting the loss curve would also add to this reliability.

**Questions:**

See questions in the 'Weakness'.

---

> ### Author Response · Authors · 2023-11-21
> **Response to Reviewer J3Ha**
>
> We thank the reviewer for the comments. Here, we provide a point-by-point response to the reviewer's comments.
>
> > **Comment:** The authors have introduced a general setting for DRO problems, including DRO with KL-Divergence and $\chi^2$-Divergence. Are there other DRO optimization problems that fit this setting?
>
> **Response:** We thank the reviewer for raising this concern. We provided two examples of DRO problems that can easily be extended to the CO setting considered in this work. In addition to the KL and $\chi^2$-Divergence, the DRO problems with Wasserstein distance can also be solved using the proposed DRO framework. We would also like to point out that even though the proposed algorithms are used to solve DRO problems they can be easily generalized to solve more general problems including, **Meta-learning**, **phase retrieval**, and **reinforcement learning**.
>
> > **Comment:** The exact form of the optimization problem in two experiments should be specified.
>
> **Response:** Thank you for the suggestion. In the revised version, we will include all the necessary details. For the experiments on the adult dataset, we have utilized BCEWithLogitsLoss(https://pytorch.org/docs/stable/generated/torch.nn.BCEWithLogitsLoss.html) with $\chi^2$-Divergence formulation presented in Sec. 2.1. While for the experiments with CIFAR-10-ST dataset, we have utilized CE loss (https://pytorch.org/docs/stable/generated/torch.nn.CrossEntropyLoss.html) with KL-Divergence formulation presented in Sec. 2.1.
>
> > **Comment:** How is the dataset distributed across each client? Does it reflect the diversity in data distribution?
>
> **Response:** We thank the reviewer for the question. Yes, the dataset is split across the nodes to capture the effect of diversity, specifically, we do not assume that each node has access to full data (homogeneous distribution). The data is split in a heterogeneous manner where each node can only see a part of the total data which naturally captures the effect of heterogeneity across different nodes.
>
> > **Comment:** Utilizing the small batch size is one strength of FedDRO. However, this isn't evident in the experiments. Specifically, |b|=16 in FedDRO, while |b|=4 (or 8) in GCIVR (with the client count m=8). Therefore, experiments should be conducted with a small batch size or more clients (ensuring a large batch size in GCIVR).
>
> **Response:** We thank the reviewer for the suggestion. In the Appendix, we have included additional experiments to compare the performance of FedDRO against other baselines with different numbers of clients. Please see Appendix B for a detailed discussion. Also, we agree that FedDRO has a major theoretical advantage in that it requires a smaller batch size compared to popular baselines. However, we believe that conducting experiments with a larger batch for GCIVR will be an unfair comparison for GCIVR since it will slow down the algorithm considerably and produce worse results than the results currently presented. However, to address the reviewer's concerns we will include additional results with larger batch sizes for the GCIVR algorithm.
>
> > **Comment:** The experiments are repeated only five times, and it would further strengthen the reliability if rerunning the experiments at least ten times. Furthermore, presenting the loss curve would also add to this reliability.
>
> **Response:** Thank you for the suggestion. In the interest of time, we were unable to rerun the experiments 10 times to average the performance. However, in the updated version we will include the experiments with additional runs.

---

### Official Review · Reviewer_Sjhd · 2023-10-29

**Soundness:** 3 good
**Presentation:** 3 good
**Contribution:** 2 fair
**Rating:** 6
**Confidence:** 3

**Summary:**

This paper proposes a new compositional optimization algorithm called FedDRO for distributionally robust learning. The proposed algorithm integrates novel communication and optimization techniques to tackle biased gradient estimates, client drift, etc. in compositional optimization. Convergence analysis and numerical experiments are conducted to validate the proposed method.

**Strengths:**

The proposed approach improves upon the previous compositional optimization methods in terms of convergence rate and the need for large batch gradients. It's also technically novel to utilize the low-dimensional communication of the compositional functions $g(\cdot)$ to trade for better convergence.

**Weaknesses:**

The low-dimensional $g(\cdot)$ still causes additional $O(\epsilon^{-2})$ communication, whose impact may vary based on specific forms of $g(\cdot)$ and the synchronization settings.

**Questions:**

N/A

---

> ### Author Response · Authors · 2023-11-21
> **Response to Reviewer Sjhd**
>
> > **Comment:** The low-dimensional $g(\cdot)$ still causes additional $O(\epsilon^{-2})$ communication, whose impact may vary based on specific forms of $g(\cdot)$ and the synchronization settings.
>
> **Response:** We thank the reviewer for the comment, we agree that in an ideal setting, we would want to avoid communication of $y_k$'s. However, as established in Section 4.2 this communication is necessary to guarantee convergence of the proposed algorithms. However, we would like to point out that, if we have $dim(g(x)) << d$, the communication requirements are manageable compared to setting when $dim(g(x))$ is large.

---

> > ### Comment · Reviewer_Sjhd · 2023-11-22
> >
> > Thanks for the response. It addresses my concerns. I'm willing to keep my score.

---

### Official Review · Reviewer_StaW · 2023-11-01

**Soundness:** 2 fair
**Presentation:** 1 poor
**Contribution:** 2 fair
**Rating:** 3
**Confidence:** 5

**Summary:**

This paper developed a federated learning method for compositional optimization problem, which can be applied to DRO problems. Both theoretical and empirical results are provided to show its performance. However, this paper overclaims its contributions. It solves a simple compositional problem so it is not surprising to see a better bound. But the authors hide this point. This could mislead the community.

**Strengths:**

This paper provides a good literature review.

Theoretical analysis is provided.

**Weaknesses:**

1. This paper overclaims its contributions.

2. The writing is poor. It could mislead the community.

3. Some operations are not practical.

4. Some claims are NOT correct.

**Questions:**

1. Different from existing federated compositional optimization, this paper investigated a simple case, i.e., the outer-level function is deterministic, not a stochastic function as the first three baselines in table 1. However, the authors never mentioned this critical difference. For this simple objective function, the authors claim they can achieve better convergence rates. This is not the case because the problem settings are different. This paper overclaims its contribution. The authors should clearly state the difference in the problem setting. Otherwise, it will mislead the community.


2. The novelty is incremental. For this simple compositional optimization problem, it is trivial to extend existing theoretical analysis to this kind of problem. I didn't see any challenges here. In particular, the outer-level function is deterministic, and $y$ is synchronized in each iteration. Then, compared with the non-compositional optimization problem, there are no additional challenges in convergence analysis.

3. For Eq.(8), what is the reason for using the storm estimator? The standard moving average estimator should also work. Could you please provide more discussions?

4. This paper claims it can achieve better communication complexity than existing works. It is NOT true. Specifically,  the proposed algorithm communicates $y$ in every iteration. Then, the communication complexity is the same as the number of iterations. It is much worse than existing approaches.

5. No experiments to compare the proposed algorithm with the federated baselines.

---

> ### Author Response · Authors · 2023-11-21
> **Response to Reviewer StaW: Part I**
>
> > **Comment:** Different from existing federated compositional optimization, this paper investigated a simple case, i.e., the outer-level function is deterministic, not a stochastic function as the first three baselines in Table 1. However, the authors never mentioned this critical difference. For this simple objective function, the authors claim they can achieve better convergence rates. This is not the case because the problem settings are different. This paper overclaims its contribution. The authors should clearly state the difference in the problem setting. Otherwise, it will mislead the community.
>
> **Response:** We thank the reviewer for the comments. We respectfully disagree with the reviewer's claim that Table 1 will mislead the community. Please allow us to clarify why the comparison presented in Table 1 is fair and why such a setting is considered in our work.
>
> First of all, we note that the stochasticity of $f(\cdot)$ can be easily handled in the same manner as that of $h(\cdot)$ considered in our formulation, and that too without making any significant changes to the presented analysis or the theoretical guarantees in the paper. Precisely speaking, we get the following guarantee when the function $f(\cdot)$ is stochastic
> $$\mathbb{E} \| \nabla \Phi(\bar{x}^{a(T)}) \|^2 \leq \mathcal{O}\bigg(\frac{1}{\sqrt{|b| K T}} \bigg) + \mathcal{O}\bigg(\frac{\sigma_g^2 + \sigma_h^2 + \sigma_f^2 }{\sqrt{|b| K T}} \bigg) + \mathcal{O}\bigg(\frac{\Delta_g^2 + \Delta_h^2 + \Delta_f^2}{\sqrt{|b| K T}} \bigg)$$
> which is similar to the one presented in Thm 5.1 but with the addition of variance and heterogeneity terms $\sigma_f^2$ and $\Delta_f^2$, respectively, corresponding to the stochasticity of the function $f(\cdot)$'s gradient evaluations. Please note that as discussed in the response to the first comment, the major challenge in developing CO algorithms comes from the inner function $g(\cdot)$ in the compositional objective which leads to biased gradients, and thereby, makes the algorithm development and analysis challenging. Note that stochasticity of $f(\cdot)$ does not present any such challenge, i.e., we can always draw a sample of $f(\cdot)$ to construct an unbiased estimate of $\nabla f(\cdot)$, exactly in the same manner as we do for $h(\cdot)$. Therefore, even for the stochastic case, the challenge comes from the distributed nature and the stochasticity of the inner function $g(\cdot)$. We considered this specific formulation since it arises in many problems of practical interest, including the DRO problems, as discussed in Sec 2 of the manuscript.
>
> In summary, the discussion corresponding to Table 1 will not change even if $f(\cdot)$ is accessed in a stochastic manner. In the revised version of the manuscript, we will include these discussions to address the reviewer's concerns.
>
> > **Comment:** The novelty is incremental. For this simple compositional optimization problem, it is trivial to extend existing theoretical analysis to this kind of problem. I didn't see any challenges here. In particular, the outer-level function is deterministic, and $y$
>  is synchronized in each iteration. Then, compared with the non-compositional optimization problem, there are no additional challenges in convergence analysis.
>
> **Response:** We thank the reviewer for the comment. We disagree with the reviewer that the analysis and the results presented in the paper are trivial.
>
> - First of all, we would like to point out that in CO problems the challenge arises from the distributed nature and stochasticity of the inner problem rather than the stochasticity of the outer problem. Also, as demonstrated by our previous response, the problem can be easily extended to the setting where the outer problem is distributed and stochastic without any major changes to the presented results. We focused on this particular class of problems because it naturally arises in many DRO problems of interest. To address the reviewer's concern in the revised version we will add a remark stating this fact.
>
> - Secondly, one of the major contributions of our work is to establish that without sharing $y$ in each iteration the proposed FedAvg-type algorithms will in fact diverge to the wrong solutions. Please see Section 4.1 of the paper for a detailed discussion of this fact and the proofs presented in Appendix D demonstrating this fact. We believe this is the first time these results have been proved and presented in the literature. In addition, we do not believe that the analysis and the results presented are trivial by any means. Please see Appendix D for more details.
>
> In summary, we believe that the analysis presented in the paper is not a trivial extension of a non-compositional problem, specifically, because in CO the major challenge in developing FL algorithms arises from the compositional nature of the inner objective that our work addresses.

---

> > ### Comment · Reviewer_StaW · 2023-11-22
> > **More questions**
> >
> > For the new convergence rate when $f$ is stochastic, the reviewer can't verify its correctness without detailed proof.
> >
> > For the communication complexity of FedAvg, it is not $O(d\epsilon^{-2})$.

---

> ### Author Response · Authors · 2023-11-21
> **Response to Reviewer StaW: Part II**
>
> > **Comment:** For Eq.(8), what is the reason for using the storm estimator? The standard moving average estimator should also work. Could you please provide more discussions?
>
> **Response:** We thank the reviewer for the comment. The standard moving average estimator will not work in this setting. The specific reason for using the STORM estimator comes from the fact that the STORM estimator allows momentum-based variance reduction and asymptotically converges to its true mean, i.e., $\bar{y}_t \to g(x_t)$ as $t \to \infty$. This behavior can be observed in Lemma F5 in Appendix F where "Descent in the estimate of g(x)" is computed. Specifically, note that how the term $\bar{y}_t \to \frac{1}{K} \sum_k g_k$ as $t$ increases. However, if we change the STORM estimator to the standard momentum estimator, we will not get any descent in $\bar{y}_t$ and we may need to rely on large batch evaluations to control the variance. We will add these discussions in the revised version of the manuscript.
>
> > **Comment:** This paper claims it can achieve better communication complexity than existing works. It is NOT true. Specifically, the proposed algorithm communicates $y$ in every iteration. Then, the communication complexity is the same as the number of iterations. It is much worse than existing approaches.
>
> **Response:** The proposed algorithm in fact achieves better communication complexity than the proposed approaches and the complexity is not worse than the existing approaches as stated by the reviewer. Please allow us to clarify.
>
> - First of all, note that $y$ is only a one-dimensional parameter which is much smaller than dimension $d$ of the model parameters. For example, even for small CNN networks with only 2 convolution layers of size $32 \times 24 \times 24$ and one fully connected forward layer of width $256$, the number of parameters is more than 200,000, i.e., $d \approx 200,000$ and sharing this $d$ is much more expensive compared to sharing $y$ with is one-dimensional. Now, note that FedDRO avoids sharing these high-dimensional parameters in each round and achieves improved communication complexity compared to the baseline algorithms which rely on sharing $d$-dimensional parameters in each iteration. Here, we demonstrate this fact.
>   - Total communication for FedDRO: $O(d \epsilon^{-3/2}) + O(\epsilon^{-2})$ vs Baseline FedAvg Algorithm: $O(d \epsilon^{-2})$. It is clear that $d \times  \epsilon^{-3/2} + \epsilon^{-2}  << d \times \epsilon^{-2}$ especially if $d$ is large. This implies that the communication requirement of FedDRO is clearly not worse than the state-of-the-art.
>
> - In addition to the communication, it is also important to note that FedDRO is the first federated CO algorithm that achieves linear speed-up with the number of clients present in the network. In contrast, none of the current approaches are capable of achieving linear speed-up with the number of clients which is an important consideration when designing FL algorithms. Please see Table 1 and the subsequent discussion.
>
> > **Comment:**  No experiments to compare the proposed algorithm with the federated baselines.
>
> **Response:** We have compared the algorithm with existing baselines for solving CO problems, the standard FedAvg algorithms will not work for the CO setting as demonstrated in Section 4 of the paper.

---

> ### Author Response · Authors · 2023-11-22
> **Response to More Questions**
>
> We thank the reviewer for the response. Here, we respond to further comments.
>
> > **Comment:** For the new convergence rate when $f$ is stochastic, the reviewer can't verify its correctness without detailed proof.
>
> **Response:** As we pointed out in the earlier response, handling the stochasticity of $f(\cdot)$ is not a major challenge in the analysis, it can be handled in a very similar manner to the stochasticity of the function $h(\cdot)$. The major challenge in the proof comes from the compositional structure and the distributed nature of the inner function $g(\cdot)$. Here, we provide highlights of where the proof will differ from the current proof to satisfy the reviewer's concern.
>
> ***Step 1:*** Lemma F3: Descent in function value will be updated to account for the variance of $f$, i.e., we will have an additional term with $\sigma_f^2$.
>
> ***Step 2:*** Lemma F4: Client Drift: The client drifts $\frac{1}{K} \sum_{k = 1}^K \\| x_k^t - \bar{x}_k^t\\|^2$ will be updated to account for the heterogeneity of the outer function $f(\cdot)$.
>
> Note that these two results are then utilized to bound the descent in the estimates of $\bar{y}^t$ and finally a very similar potential function as used in the current analysis yields the final result of the theorem as stated in the above response.
>
> We would like to again emphasize that the stochasticity of $f(\cdot)$ can be handled exactly in the same manner as that of the function $h(\cdot)$ in the current analysis and does not pose any additional challenges compared to the current analysis of the paper.
>
> > **Comment:** For the communication complexity of FedAvg, it is not $O(d \epsilon^{-2})$.
>
> **Response:** It is well understood and known that the communication complexity of FedAvg is $O(\epsilon^{-2})$ please see [1]. This implies that the total communication for FedAvg will be $O(d \epsilon^{-2})$ as stated in our previous response. We would like to also point out that some papers have even shown communication complexity of  $O(\epsilon^{-3/2})$ for FedAvg [2] for such settings the total communication will be $O(d \epsilon^{-3/2})$ which is comparable to our setting, especially for $d >> 1$.
>
> However, when we made the above statement we were not directly comparing our algorithm to FedAvg since FedAvg is an algorithm that is designed to solve non-composite problems. We just wanted to bring to the reviewer's attention that the complexity is not much worse than the existing approaches as claimed by the reviewer.
>
> In addition, we would like to add that our algorithm not only improves previous results but also achieves linear speed-up with the number of clients present in the network. We would like to note that this is certainly a desirable property for distributed algorithms which the current algorithms are incapable of achieving.
>
>
> [1] Karimireddy, Sai Praneeth, et al. "Scaffold: Stochastic controlled averaging for federated learning." International conference on machine learning. PMLR, 2020.
>
> [2] Yu, Hao, Sen Yang, and Shenghuo Zhu. "Parallel restarted SGD with faster convergence and less communication: Demystifying why model averaging works for deep learning." Proceedings of the AAAI Conference on Artificial Intelligence. Vol. 33. No. 01. 2019.
>
> If you have any further concerns please feel free to post them. We will be happy to address them.

---

### Official Review · Reviewer_EdtR · 2023-11-10

**Soundness:** 2 fair
**Presentation:** 3 good
**Contribution:** 2 fair
**Rating:** 5
**Confidence:** 4

**Summary:**

This paper develops a momentum-type federated learning algorithm for Compositional Optimization (CO) problems. The authors provide an $O(\epsilon^{-2})$ sample complexity and $O(\epsilon^{-3/2})$ communication complexity for their approach. Numerical experiments on large-scale Distributionally Robust Optimization problems demonstrate the effectiveness of their method.

**Strengths:**

1. The paper addresses an important problem in the field.

2. The paper is well-written, with clear mathematical notations. The table comparing related works provides insightful information on their contributions. The discussions around each assumption are useful.

3. The discussion surrounding the theorems is useful for understanding their results.

**Weaknesses:**

Please see below.

**Questions:**

**Abstract:**

- The following expression has already been discussed in existing literature. See Lemma 2.1 of [R1] and Section 1.2 of [R2]. What are your findings in contrast to these?

    “We first establish that vanilla FedAvg is not suitable for solving distributed CO problems due to data heterogeneity in the compositional objectives at each client. This leads to the amplification of bias in the local compositional gradient estimates.”

- The following sample and communication complexity for CO problems have already been established. See [R3] and [R4] for details on sample and communication complexity, and [R4] for linear speedup under the heterogeneity assumption (Assumption 3.4 in your paper). Please clarify the differences.

    “We establish an $O(\epsilon^{-2})$ sample and $O(\epsilon^{-3/2})$ communication complexity in the FL setting while achieving linear speedup with the number of clients.”

**Section 4 (Algorithm Design):**

Algorithm 1 is similar to the method proposed in [R3] and [R4]. Specifically, the improved communication complexity and sample complexity are obtained from Eq (7) and the momentum update in Eq (8), which are already studied in [R3] and [R4].

**Experiment:**

- The main focus of the paper is federated learning under the heterogeneity assumption. However, this setting is not apparent in the experiment evaluation.


- It would be useful if the authors provided a comparison to variance reduction and momentum-based methods designed for heterogeneous federated composition problems [R1-R4].

- Why is there a jump in test accuracy in Figure 1 after 100 communications?



Please let me know in your response if I misunderstood your contribution, and I will be happy to update my score.




[R1] Tarzanagh, D.A., Li, M., Thrampoulidis, C. and Oymak, S., 2022, June. Fednest: Federated bilevel, minimax, and compositional optimization. In International Conference on Machine Learning (pp. 21146-21179). PMLR.

[R2] Yang, S., Zhang, X. and Wang, M., 2022. Decentralized gossip-based stochastic bilevel optimization over communication networks. Advances in Neural Information Processing Systems, 35, pp.238-252.


[R3] Feihu Huang. Faster adaptive momentum-based federated methods for distributed composition optimization. arXiv preprint arXiv:2211.01883, 2022

[R4] Tarzanagh, D.A., Li, M., Sharma, P. and Oymak, S., 2023. Federated Multi-Sequence Stochastic Approximation with Local Hypergradient Estimation. arXiv preprint arXiv:2306.01648.

---

> ### Author Response · Authors · 2023-11-22
> **Response to Reviewer EdtR: Part I**
>
> > **Comment:** The following sample and communication complexity for CO problems have already been established. See [R3] and [R4] for details on sample and communication complexity, and [R4] for linear speedup under the heterogeneity assumption (Assumption 3.4 in your paper). Please clarify.
>
> **Response:** We thank the reviewer for the insightful comments. Here, we contrast our results and algorithms compared to [R3] and [R4].
>
> - **Comparison to [R3]:** We would like to bring to the reviewer's attention that the problem considered in [R3] is different from the problem considered in our work. The authors in [R3] consider the objective function
>   $$\frac{1}{k}\sum_{k = 1}^K f_k(g_k(\cdot)) \qquad (1),$$
> please observe that in this setting the local nodes have access to local composite functions $f_k(g_k(\cdot))$. In contrast, we consider a setting where the objective function is
> $$\sum_{k = 1}^K h_k(x) +  f\bigg(\frac{1}{k} \sum_{k=1}^K g_k(\cdot)\bigg) \qquad (2),$$
> where the local nodes have access to only $h_k(\cdot)$ and $g_k(\cdot)$. Note that the major difference in the two settings in eq. (1) and (2) come from the fact that in eq. (1) the inner function $g_k(\cdot)$ is fully available at each node, whereas in eq. (2) the inner function $1/K \sum_{k = 1}^K g_k(\cdot)$ is not available (since each node can only access $g_k(\cdot)$) at the local nodes.
>
>   Importantly, we point out that the setting in eq. (2) is more practical as can be seen from the examples presented in Section 2.1 wherein the DRO problems take the form of eq. (2) rather than eq. (1) in a distributed setting. Next, we show why the algorithms developed for [R3] cannot be utilized to solve the problem considered in our work.
>
>    - **Challenges in solving eq. (2):** A major contribution of our work is in establishing the fact that the algorithms that are developed for solving eq. (1), i.e., the algorithms developed in [R3], cannot be utilized to solve the problem considered in our work.
>
>       To demonstrate this consider the simple deterministic setting with $f_k = f$, then the local gradient computed for the objective function in eq. (1) will be $\nabla g_k(x) \nabla f(g_k(x))$. Note that this is an unbiased local gradient which further implies that simple FedAVG-based implementations can be developed for solving this problem as done in [R3]. In contrast, note that the local gradient $\nabla g_k(x) \nabla f(g_k(x))$ will be a biased local gradient for our problem in eq. (2) and will lead to the divergence of FedAvg-based algorithms [R3] as shown in Section 4.1. Moreover, note that we establish that even if we share the local functions $g_k(\cdot)$ intermittently among nodes we may not be able to mitigate the bias of local gradient, and the developed algorithms will diverge to incorrect solutions. Please see Section 4.1 for more details.
>
>
> - **Comparison to [R4]:**  We thank the reviewer for pointing us to [R4]. We agree with the reviewer that [R4] achieves similar computation complexity including linear speed-up as in our algorithm. However, we achieve a better communication complexity compared to [R4]. In addition to the proposed algorithm, one of the novel contributions of this work is in Section 4.1 where we have theoretically and experimentally established that vanilla FedAvg-type algorithms will not work for solving CO problems unless some additional (low-dimensional) communication is performed. Based on this analysis, we would like to make a more important point regarding the algorithm considered in [R4].
>
>    - If we take a detailed look at the proofs of Appendix D, it can be clearly seen that the algorithm proposed in [R4] also cannot solve the CO problem considered in our work. Below, we justify our reasoning.
>
>        We have established that for a setting with $K = 2$ nodes with $g_1(x) = 4x - 4$ and $g_2(x) = -2x + 4$ and function $f$ such that
>          $$f (g(x)) =  f \Big(\frac{1}{2} (g_1(x) + g_2(x))\Big)  = \sqrt{\Big[ \frac{1}{2} (g_1(x) + g_2(x)) \Big]^2}. $$
>        For this problem, if we have access to $g_1$ and $g_2$ in a deterministic form along with the exact deterministic local composite gradient, any FedAvg-type algorithm will diverge. Note that the algorithm FedMSA in [R4] considers local momentum-based stochastic estimates for both $g_1$ and $g_2$ and the outer function $f$ which are strictly worse than the deterministic estimates constructed in Appendix D. This implies that FedMSA cannot solve the CO problem of our paper$^1$.
>
> In summary, we have established that the problem considered in [R3] is different than the one developed in [R4], in addition, we have theoretically established that the algorithms developed in [R4] cannot be utilized to solve the problems considered in our work.
>
>
> $^1$ We believe this observation leads to a contradiction that requires a more detailed analysis of the algorithms and assumptions utilized in [R4] to guarantee convergence.

---

> ### Author Response · Authors · 2023-11-22
> **Response to Reviewer EdtR: Part II**
>
> > **Comment:** Algorithm 1 is similar to the method proposed in [R3] and [R4]. Specifically, the improved communication complexity and sample complexity...
>
> **Response:** Please see the response above for a detailed comparison with [R3] and [R4]. One additional thing we would like to point out is that in [R4] the local function estimates and gradients are both updated using a momentum estimator, however, in our setting only $y$ is updated using a momentum-based estimator which is different from that of [R4], moreover the local gradients are estimated using popular SGD like estimates.
>
> > **Comment:** The following expression has already been discussed in existing literature. See Lemma 2.1 of [R1] and Section 1.2 of [R2]. What are your findings in contrast to these?
> >
> > “We first establish that vanilla FedAvg is not suitable for solving ...”
>
> **Response:** We agree with the reviewer that the above statement has been discussed in the past in [R1] and [R2] in some form or another in different settings. However, there are a few differences between the reasoning and the motive of the above statement in our work compared to [R1] and [R2]. Below, we clarify our reasoning and contrast it with [R1] and [R2].
>
> - Both the works [R1] and [R2] state that since the inner function (or $y^\ast(x)$ for bilevel optimization) is not available globally the local gradients will have some bias and vanilla algorithms may not work for the CO problem. However, they do not establish that even if the (exact) inner function (or $y^\ast(x)$ for the bilevel problem) is made available at each node in each communication round the algorithms may still diverge. This is exactly what we establish in our work and these results/discussions are missing in [R1] and [R2]. Specifically, our work is the first to establish both theoretically and empirically that even if we allow the sharing of the inner functions intermittently, the vanilla FedAvg algorithm will not work (even with only two local updates) and will diverge to the wrong solutions. Please see Case II of Algorithm 1 in the paper. Note that these results have never been observed or discussed in the literature in the past.  We would also like to point out that these negative results are significant since they allow a better understanding of the effect of local updates and may lead to the development of better algorithms in the future for solving CO problems.
>
> - In addition, we would also like to point out that the algorithms developed in [R1] and [R2] are significantly different from the ones developed in our work. First, the algorithm developed in [R1] is a bi-level algorithm with a multi-loop structure with many tunable (hyper) parameters. Such algorithms are not preferred in practical implementations. In contrast, our algorithm
> is a single-loop algorithm with simple FedAvg-type SGD updates. In addition to being practical,
> our work also significantly improves upon the theoretical guarantees achieved by [R1] by achieving linear speed-up with the number of clients as well as improved communication complexity. Secondly, [R2] considers a decentralized setting with no local updates where each component of the (partial) gradient is updated using momentum estimators.
>
> **Experiments:**
>
> > **Comment:** The main focus of the paper is federated learning under the heterogeneity assumption. However, this setting is not apparent in the experiment evaluation.
>
> **Response:** We thank the reviewer for the comment. The experiments are conducted with heterogeneous data splits across each node. Specifically, the data is split among nodes in a manner where each node observes a disjoint set of data points hence modeling the realistic heterogeneous settings. In the revised version of the paper, we will clarify the experiment settings and add more experiments with heterogeneous settings.
>
> > **Comment:** It would be useful if the authors provided a comparison to variance reduction and momentum-based methods designed for heterogeneous federated composition problems [R1-R4].
>
> **Response:** As discussed earlier, the problem setting considered in [R3] is different than the one considered in our work and the algorithm in [R4] is incapable of solving the problems considered in our work. To address the reviewer's concern, we plan to compare our algorithm with the algorithms proposed in [R1] and [R2], however, note that since [R1] is a multi-loop algorithm with no linear speedup and [R2] relies on model/gradient sharing at each iteration, we can expect the performance of these algorithms to be significantly worse compared to the proposed FedDRO which is both communication and computationally efficient (because of linear speed up).
>
> > **Comment:** Why is there a jump in test accuracy in Figure 1 after 100 communications?
>
> **Response:** After 90 communication rounds, we reduce the learning rate to $1/10$ which leads to a jump in the test accuracy. We have discussed this in the experiments.

---

> ### Comment · Reviewer_EdtR · 2023-11-23
> **Response to Authors**
>
> Dear Authors,
>
> I appreciate your taking the time to respond to my comments.
>
> > The following expression has already been discussed in existing literature. See Lemma 2.1 of [R1] and Section 1.2 of [R2]. What are your findings in contrast to these?
>
> Addressed.
>
> > The following sample and communication complexity for CO problems have already been established. See [R3] and [R4] for details on sample and communication complexity, and [R4] for linear speedup under the heterogeneity assumption (Assumption 3.4 in your paper). Please clarify the differences.
>
>
> Not Addressed.
>
> * I respectfully disagree with  your response that "we achieve a better communication complexity compared to [R4]." Could you please clarify the communication complexity in comparison to [R4]? I believe the communication complexity in [R4], i.e., $O(\tau \epsilon^{-1})$, is already near-optimal, where $\tau$ is the heterogeneity parameter.
>
> * I agree that the standard FedAvg may not be an appropriate method (i.e., with theoretical guarantees) for solving the federated CO problems in your work and in [R4]. However, client drift reduction methods with one additional communication in each round and under certain heterogeneity assumptions can provide some guarantees, as established in [R4].
>
>
> > Algorithm 1 is similar to the method proposed in [R3] and [R4]. Specifically, the improved communication complexity and sample complexity are obtained from Eq (7) and the momentum update in Eq (8), which are already studied in [R3] and [R4].
>
>
> Addressed.
>
> It would be good if the authors discussed the similarities and differences in the main paper.
>
> > The main focus of the paper is federated learning under the heterogeneity assumption. However, this setting is not apparent in the experiment evaluation.
>
>
> Not Addressed.
>
> I don't think the phrase “the data is split among nodes… realistic heterogeneous settings” adequately explains federated learning under heterogeneity. My suggestion is to provide a better heterogeneity federated learning setting and a parameter to control it, so that you can observe the impact of heterogeneity on convergence and communication. See, for example [R4].
>
>
> > It would be useful if the authors provided a comparison to variance reduction and momentum-based methods designed for heterogeneous federated composition problems [R1-R4].
>
> Not Addressed.
>
> I respectfully disagree with your response, “ [R4] is incapable of solving the problems considered in our work,” as
> [R4] considers the drift reduction/variance reduction method, not the standard FedAvg.
>
> > Why is there a jump in test accuracy in Figure 1 after 100 communications?
>
> Addressed.

---

> ### Author Response · Authors · 2023-11-23
> **Further Response to Reviewer's Concerns**
>
> We thank the reviewer for the insightful suggestions. Here, we attempt to address the remaining concerns of the reviewer.
>
> > **Comment:** I respectfully disagree with your response that "we achieve a better communication complexity compared to [R4]." Could you please clarify the communication complexity in comparison to [R4]? I believe the communication complexity in [R4], i.e., $O(\tau \epsilon^{-1})$, is already near-optimal.
>
> >  I agree that the standard FedAvg may not be an appropriate method (i.e., with theoretical guarantees) for solving the federated CO problems in your work and in [R4]. However, client drift reduction methods with one additional communication in each round and under certain heterogeneity assumptions can provide some guarantees, as established in [R4].
>
> **Response:** Thank you for pointing this out. We agree with the reviewer that [R4] achieves better communication complexity compared to our approach. Let us again clarify the point we wanted to make in our previous response. We believe that there are potential mistakes in the proof of the results presented in [R4], specifically, we believe that the algorithm presented in [R4] cannot converge in theory. Below, we justify the reasoning behind our belief.
>
> - If we take a detailed look at the proofs in Appendix D of our manuscript, we can see that for the example with $K = 2$ nodes and with inner functions $g_1$ and $g_2$ defined as: $g_1(x) = 4x - 4$ and $g_2(x) = -2x + 4$ and function $f$ defined as
>
>      $$f (g(x)) =  f \Big(\frac{1}{2} (g_1(x) + g_2(x))\Big)  = \sqrt{\Big[ \frac{1}{2} (g_1(x) + g_2(x)) \Big]^2 + 4}. $$
>
>     For this problem, if we have access to $g_1$ and $g_2$ in a deterministic form along with the exact deterministic local composite gradient, any algorithm that performs local updates with intermittent sharing (FedAvg-type algorithms) of $y$ and $x$ will diverge, i.e., converge to wrong solutions. Please see Section 4.1 along with Case I and Case II in Algorithm 1 of our manuscript.
>
> - Now let us try to solve the same problem with the algorithm proposed in [R4]. Note that since the inner function $g_i$'s for i = \{1,2\} and the outer function $f$ are deterministic, the variance reduced estimators proposed in equations (5a) and (5a) are exactly the full local gradient estimators, i.e., we have using (5a) and (5b) that
>
>      $$\text{Local grad. at node 1} =  \nabla g_1(x) \nabla f( g_1(x) ) = 4 \frac{ y_1}{\sqrt{y_1^2 + 4}},  ~~\text{Local grad. at node 2} =  \nabla g_2(x) \nabla f( g_2(x) ) = - 2 \frac{ y_2}{\sqrt{y_2^2 + 4}}.$$
>
>     Note that with these local gradient updates and the intermittent sharing of sequences $y$ (local estimates of $g_i$'s) and $x$ (model parameters), as proposed in [R4], Algorithm 1 in [R4] and Case II of Algorithm 1 in our paper are exactly same. This implies that the algorithm in [R4] should not be able to solve the CO problem discussed in Appendix D as established in Section 4.1 of our paper. This makes us believe that the results presented in [R4] may have some errors. However, given the time limitation, we have not been able to pinpoint the exact source of error in [R4], but we believe our proofs contradict the results presented in [R4].
>
> -  We would also like to point out that [R4] relies on additional assumptions of mean square Lipschitzness of the function $f$ (and $h$) which allows for variance reduction however, we do not need this assumption.
>
>
> > **Comment:** I respectfully disagree with your response, “ [R4] is incapable of solving the problems considered in our work,” as [R4] considers the drift reduction/variance reduction method, not the standard FedAvg.
>
> **Response:** When we stated that "[R4] is incapable of solving the problems considered in our work", we wanted to say that Algorithm 1 in [R4] is exactly the same as Case II of Algorithm 1 presented in the manuscript for a deterministic setting.  And as we have established Case II of Algorithm 1 cannot converge to a correct solution unless some form of information sharing is performed at each iteration. Please see the above response for more details.
>
> > **Comment:** I don't think the phrase “the data is split among nodes… realistic heterogeneous settings” adequately explains federated learning under heterogeneity. My suggestion is to provide a better heterogeneity federated learning setting and a parameter to control it so that you can observe the impact of heterogeneity on convergence and communication. See, for example [R4].
>
> **Response:** We appreciate the reviewer's suggestion. In the current version of the experiments, we do not vary the heterogeneity of the data. The current plots and results are generated under natural heterogeneity incurred by standard data split. As suggested by the reviewer, in the revised version, we will allow the system to control the data heterogeneity by allowing each node to access only a subset of classes. We hope this will address the reviewer's concern.